# Enhancing Social Intelligence in LLMs with Hierarchical Reasoning and Utterance-Level Goal Rewarding

## Abstract

Large language models (LLMs) excel in structured tasks but struggle with dynamic social interactions, where success requires long-term goal coordination and rapid adaptation. Current methods often apply uniform goal-based rewards to every utterance, overlooking the specificity of objectives at each dialogue turn and failing to account for the rationale of potential strategies. Inspired by the Theory of Planned Behavior, we propose the Think-Strategy-Response **(TSR)** framework, which decomposes social dialogue into two hierarchical stages: high-level strategic planning and low-level linguistic execution. To optimize TSR, we introduce Linearized Hierarchical Reinforcement Learning with Variance-Gated Rewards **(LHRL-VGR)**, a novel algorithm that dynamically routes rewards—balancing goal completion and strategy adherence—based on the variance of goal achievement scores. Experiments on the SOTOPIA benchmark show that our approach fine-tunes a Qwen2.5-7B agent to surpass the GPT-4o baseline by 7.32% in goal completion success, demonstrating state-of-the-art performance in multi-agent social negotiation tasks.

## 1 Introduction

Large language models (LLMs) perform well on static tasks with clear rules and a single correct answer, such as mathematics, programming, and formal logic (Yang et al., 2024; DeepMind, 2024; Jaech et al., 2024; Guo et al., 2025; OpenAI, 2025). Yet the reasoning these tasks require is quite different from the reasoning demanded in complex social interactions—especially negotiations with conflicting interests and agents pursuing long-term goals. By simulating human behavior, LLM-based agents create new opportunities to train strategic social reasoning and to explore dynamic, multi-agent environments (Wang et al., 2025d). However, LLMs struggle in these dynamic social settings. To succeed, they need to coordinate long-term goals and adapt quickly, which remains a major challenge for them (Zhou et al., 2024; Zhang et al., 2024; Liu et al., 2025a).

Numerous studies have indicated that in social dialogues, utterances are typically underpinned by varying degrees of cognitive processes and guided by latent goals (Sperber & Wilson, 1986; Levinson, 2019). Currently Most optimization methods focus on goal-oriented training. This includes both end-to-end approaches and plug-and-play planners, evolving from episode-level to more detailed utterance-level training methods (Deng et al., 2024; Liu et al., 2025a; Zhang et al., 2025). Wang et al. (2025b) demonstrated that incorporating multiple diverse chains of thought can enhance performance. Although these approaches show promise, two critical questions remain: (1) Are the agents truly learning human-like social inference, or are they merely identifying shortcut utterances that maximize success? (2) Does applying uniform *goal-based* rewards to every utterance overlook the specificity of *objectives* at each dialogue turn?

Motivated by these questions, we seek a more principled approach. Guided by the Theory of Planned Behavior (TPB) (Conner, 2020), this study delineates the simulation of human social intelligence into three core components: **Thinking** is shaped by attitudes toward the behavior, perceived social norms, and beliefs about control (Ajzen, 1991); **Strategy** is the behavioral intention formed from this cognitive evaluation (Armitage & Conner, 2001); **Action** is the observable behavior that manifests when intention aligns with opportunity and actual behavioral control (Webb & Sheeran, 2006).

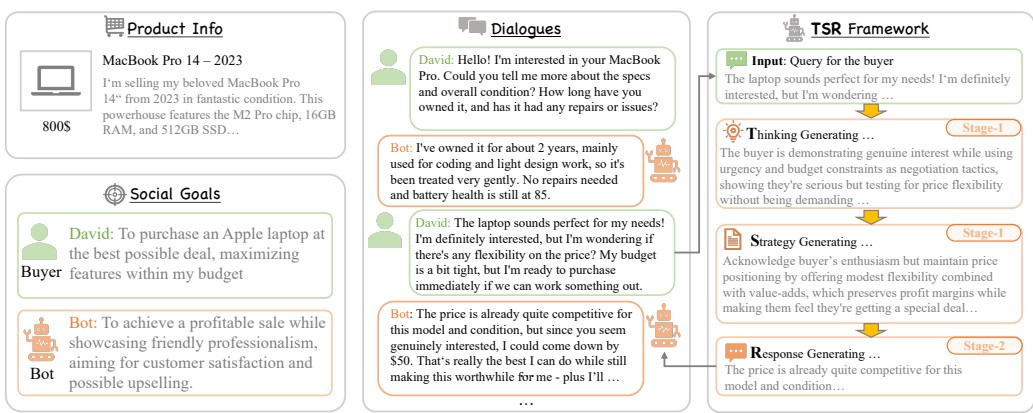

Figure 1: An exemplary scenario demonstrating the TSR framework. The social task in this work involves a given scenario and predefined social goals for both participants, who engage in a multi-turn dialogue to maximize the achievement of their respective goals in left.

To enable LLM agents to emulate this social behavior pattern, we formulate a two-stage generation framework termed the **T**hink-**S**trategy-**R**esponse (**TSR**) framework. As illustrated in Figure 1, the framework decomposes social tasks into two sequential phases. This decomposition directly addresses our second question by ensuring that each dialogue turn is guided by a specific, high-level strategy, thus moving beyond the application of a uniform goal reward to every utterance.

Inspired by the abstraction principle in Hierarchical Reinforcement Learning (HRL) (Pateria et al., 2021), which employs a high-level policy to set subgoals and a low-level policy to execute actions, we map the strategic planning and linguistic execution in social reasoning to this hierarchical structure. However, to adapt to the turn-based nature of dialogue, we linearize the process into a sequential, per-turn generation pipeline.

As a result, to operationalize this framework and tackle the first question—ensuring agents learn genuine social inference rather than shortcut utterances—we introduce **L**inearized **H**ierarchical **R**einforcement **L**earning with **V**ariance-**G**ated **R**ewards (**LHRL-VGR**). The key innovation is a novel reward structure: the high-level strategy generator is optimized with a composite reward that values foresight and strategic coherence, while the low-level response generator is trained with a dynamic, variance-gated reward. This mechanism intelligently routes rewards, prioritizing strategy adherence when goal achievement scores are stable and goal completion when they are uncertain. This approach ensures that response generation is not myopically focused on short-term goal success but is disciplined by a coherent strategy, thereby fostering human-like social reasoning. Experimental results demonstrate that our method enables the TSR agent to achieve state-of-the-art performance against strong baselines.

The main contributions of this paper are summarized as follows:

- We propose the Think-Strategy-Response (TSR) framework, a linearized hierarchical paradigm that explicitly decomposes social reasoning into strategic planning and linguistic execution, enabling more rational and interpretable dialogue generation.

- We develop LHRL-VGR, a novel algorithm that introduces a variance-gated reward router to dynamically assign goal-completion or strategy-adherence rewards to the response generator, while jointly training both policy levels through a composite strategic reward.

- Our method achieves state-of-the-art performance on the SOTOPIA benchmark, where a Qwen2.5-7B agent fine-tuned with LHRL-VGR outperforms the powerful GPT-4o baseline by **7.32%** on average goal completion success.

## 2 RELATED WORK

**Social Intelligence.** Social intelligence (Bandura et al., 1986; Kihlstrom & Cantor, 2000; Gardner, 2011) involves understanding states, intentions and behaviors of human, and navigating complex social interactions. Since the LLM like ChatGPT has shown great promise in understanding human intentions, researchers around world try to utilize reinforcement learning (RL) to enhance the social capabilities of LLMs. EPO (Liu et al., 2025b) enhances complex and strategic reasoning capabilities of LLMs through reinforcement learning (RL) techniques. AMPO (Wang et al., 2025a) improves social reasoning by dynamically selecting among different thinking models. AMPO refines the advantage function to optimize thinking mode switching module. However, both EPO and AMPO primarily focus on the strategy-level rather than utterance-level, which may lead to unstable and bad user experience. SOTOPIA-RL (Yu et al., 2025) is a RL framework to advance social intelligence in LLMs. SOTOPIA-RL focuses on the utterance-level optimization and introduces multi-dimensional rewards to enhance the social capability. Nevertheless, utterance-level rewards in SOTOPIA-RL only consider single-turn scenarios and fails to capture differs of social objectives.

**Reinforcement Learning for LLMs.** Recent advances in LLMs have demonstrated the effectiveness of RL in complex tasks. CRF (Zhang & Zhao, 2025) integrates RL and LLMs through a novel hierarchical agent architecture for question answering, addressing hallucination issues in scenarios requiring long reasoning. ReMA (Wan et al., 2025) is a hierarchical multi-agent RL framework, enabling LLMs to develop meta-cognitive abilities through both high-level and low-level reasoning agents. In social intelligence, EPO (Liu et al., 2025b) employs a dedicated strategic reasoning LLM to generate real-time strategies for arbitrary LLM agents in interactive environments. Although EPO pays attention to the strategy, EPO lacks consideration of the consistency between the result and the strategy, leading to potential hallucinations. In contrast, we propose the LRVR-VGR framework, which are designed to optimize both strategies and actions at the same time.

## 3 TSR (THINK-STRATEGY-RESPONSE) GENERATION FRAMEWORK

In this paper, each social task is defined by a specific scenario involving two social agents, each with distinct role profiles and private social goals. Two role-playing social agents, denoted as $\pi_{\theta_1}$ and $\pi_{\theta_2}$, engaging in a sequential dialogue interaction. The response generated by a social agent at turn $i$ is formulated as:

$$a_i^{\pi_{\theta_j}} \sim \pi_{\theta_j}(\cdot \mid s, p, g, h_{1:i-1}), \quad j \in \{1, 2\}, \tag{1}$$

where $s$ denotes the scenario, $p$ denotes the persona of the current speaking agent, $g$ represents its private social goal, and $h_{1:i-1} = \{a_1, a_2, a_3, \ldots, a_{i-1}\}$ captures the interaction history up to turn $i - 1$. However, this formulation does not explicitly consider the *strategic reasoning processes* involved in aligning with long-term objectives, thereby overlooking many cognitive processes in communication that remain unarticulated in overt discourse.

To address this, inspired by the strategic reasoning proposed in EPO (Liu et al., 2025b) and the Theory of Planned Behavior (TPB) (Conner, 2020), we propose a two-stage pipeline as shown in the rightmost column of Figure 1. In the first stage, to better simulate human reasoning processes, we use prompt engineering to guide the strategy model $\pi_{\text{strategy}}$ in emulating human-like **thinking**—including speculation, inference, analysis, and summarization—which forms the cognitive basis (attitudes, perceived social norms, and control beliefs) for decision-making. Following this thinking phase, a **strategy** (behavioral intention) is generated. This strategy provides *high-level guidance* for subsequent response, translating the cognitive evaluation into an actionable plan and enhancing contextual coherence. Grounded in the principle that human decision-making inherently incorporates outcome anticipation (Rangel et al., 2008), the strategy generation also integrates the *expected objectives* of actions to reflect current operational goals.

In the second stage, the prompt directs the action model $\pi_{\text{response}}$ to generate **response** based on the strategy from the first stage. Recent studies suggest that excessive reasoning content can impair performance. Since our strategy is already a highly refined product of prior reasoning, the first stage's thinking content is not included in the second stage's input, nor is any additional reasoning process introduced. This design enhances the efficient use of output information, reduces computational overhead, and aligns with the hierarchical nature of human social reasoning.

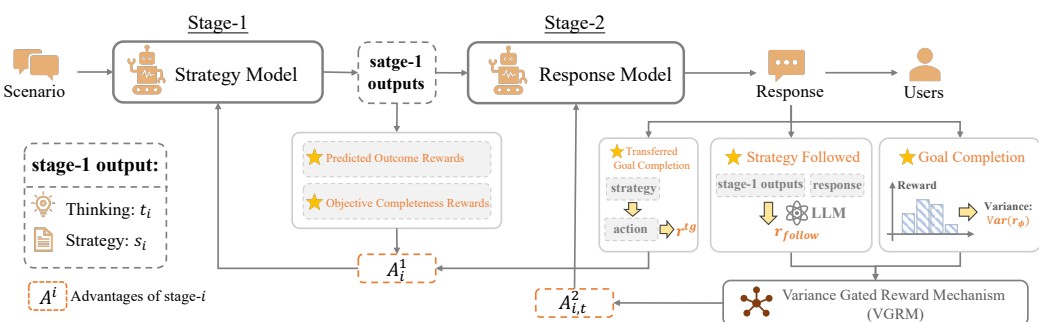

Figure 2: The pipline of LHRL-VGR.

Therefore, at turn i our framework follows the following output pipeline:

$$t_i, s_i \sim \pi_{\text{strategy}}(\cdot \mid s_{\text{strategy}}, p, g, h_{1:i-1}), \tag{2}$$

$$a_i \sim \pi_{\text{response}}(\cdot \mid s_{\text{response}}, p, s_i, g, h_{1:i-1}). \tag{3}$$

where $s_{\text{strategy}}$ and $s_{\text{response}}$ denote the prompt contents for the two stages; see Appendix J.1 for details. This two-stage decomposition enables models to concentrate on specific components of generation. It also facilitates strategy-targeted supervision and the design of more flexible, mode-conditioned action rewards. Some examples of TSR generation can be found in the Appendix I. To reduce computational overhead and simplify downstream training, both stages in this work employ the same policy model.

## 4 LHRL-VGR

### 4.1 REWARD SETTINGS

During the reinforcement learning process of our TSR framework, the agent is guided by a composite reward signal in the first stage and a dynamic reward signal in the second stage. Next, we provide a detailed explanation of the formulation and allocation of these rewards.

#### 4.1.1 REWARD FOR RESPONSE

We designed two distinct reward functions, each corresponding to a different mode (as defined in Section 3): the Strategy-followed Reward (social-oriented) and the Goal-completion Reward (goal-oriented).

**Goal-completion reward** ($r^{\text{goal}}$). The turn-level goal-completion reward evaluates how well the response improves the completion of the goal. Following recent work (Deng et al., 2024; He et al., 2024; Liu et al., 2025a), we implement a robust LLM evaluator $r_\phi(\cdot)$ to assess the progress of goal completion at each turn. The evaluator assigns a score in the range $[0, 10]$, where 0 indicates no progress and 10 represents complete achievement of the goal. For each answer $a_i$, the reward is computed based on the difference $g_i$ between the goal completion scores before and after the response. To ensure training stability, we adopt the boundary-aware scaling function from AMPO (Wang et al., 2025b), which dynamically adjusts the magnitude of difference based on the distance from the current score to the boundaries while mapping the scaled difference to the $[0, 1]$ interval through a linear transformation:

$$r^{\text{goal}} = \frac{\hat{g}_i + 1}{2}, \quad \hat{g}_i = \begin{cases} \dfrac{g_i}{10 - s_t}, & \text{if } g_i \geq 0 \\ \dfrac{g_i}{s_t}, & \text{if } g_i < 0 \end{cases} \tag{4}$$

where $\hat{g}_i \in [-1, 1]$ is boundary-aware scaling function. $g_i = r_\phi(s_t, a_i) - s_t$ is the raw difference, $s_t$ is the goal completion score before response at turn $t$, $r_\phi(s_t, a_i)$ is the score after response $a_i$.

**Strategy-followed reward** ($r^{\text{follow}}$). In prior research on hierarchical reinforcement learning, the high-level policy typically sets a subgoal for the low-level policy and provides corresponding re-

wards based on goal achievement (Pateria et al., 2021; Vezhnevets et al., 2017). Inspired by this approach, we design a reward mechanism for responses that evaluates their degree of policy alignment. By establishing strict criteria, we employ another LLM evaluator $r_\gamma(s_t, a_i)$ to perform listwise scoring of $k$ generated responses on a scale from 1 to 5:

$$r^{\text{follow}} = r_\gamma(\text{strategy}, a_i) \tag{5}$$

**Non-repetition Reward** ($r^{\text{rep}}$). In multi-turn interactions, the model can repeat content from its previous responses, which reduces informational value and inflates context length. To discourage such repetition and encourage novelty, we penalize bigram overlap with the model's prior turn using a ROUGE-2 F1 (Lin, 2004) based reward:

$$r^{\text{rep}} = \begin{cases} 1, & \text{if } i \in \{0, 1\}, \\ 1 - \text{R2}(a_i, a_{i-2}), & \text{if } i \geq 2, \end{cases} \tag{6}$$

where $a_i$ is the model's answer at turn $i$ and $a_{i-2}$ is its previous answer at turn $i-2$ (under standard user–assistant alternation). Since $\text{R2}(a_i, a_{i-2}) \in [0, 1]$, the reward $r_i^{nr} \in [0, 1]$, with higher values indicating less bigram overlap (greater novelty) and lower values indicating stronger repetition of the prior response.

**Reward routing mechanism.** A key research question is how to balance the goal-completion reward and the strategy-followed reward more effectively. We argue that the goal-completion reward should still serve as the core metric, and the strategy-followed reward should be considered for adjustment only when the variation in utterance goal completion is largely determined. It is important to note that during the calculation of the Goal-completion reward, we employ an LLM evaluator $r_\phi(\cdot)$ to estimate the current goal completion score. However, due to *inherent biases* in large language models, the scores assigned by the LLM may vary even for nearly identical dialogue contents, as discussed in Appendix D. Therefore, to better select between the goal-completion reward and the strategy-followed reward, for each instance, the variance in goal completion scores across its multiple response samples is used as the criterion (**VGRM** in Figure 2). If the variance exceeds a predefined threshold $\theta$, the Goal-completion reward is calculated; otherwise, the Strategy-followed reward is applied, the final reward for response is expressed below:

$$r^{\text{res}} = \begin{cases} r^{\text{goal}} \cdot r^{\text{rep}}, & \text{if } \text{Var}\big(r_\phi(s, a_1), r_\phi(s, a_2), \ldots, r_\phi(s, a_k)\big) > \theta \\ r^{\text{follow}} \cdot r^{\text{rep}}, & \text{if } \text{Var}\big(r_\phi(s, a_1), r_\phi(s, a_2), \ldots, r_\phi(s, a_k)\big) \leq \theta. \end{cases} \tag{7}$$

To validate the rationale behind our routing approach, we also implemented two alternative routing strategies for comparison. The specific configurations and corresponding results are presented in Appendix C.

### 4.1.2 REWARD FOR STRATEGY

In Section 3, we define our strategy to comprise two components: behavior guidance and utterance-level objectives. Accordingly, we design two distinct rewards: a predicted outcome reward and a content completeness reward.

**Reward for the strategy content** ($r^s$). (1) Predicted outcome reward ($r^{\text{predict}}$). To ensure alignment with the ultimate goal, this reward involves predicting the behaviors of both parties after executing each proposed strategy. A score between 0 and 5 is assigned based on the estimated outcome, primarily reflecting the extent to which the strategy contributes to effective progress toward the goal. (2) Content completeness reward ($r^{\text{complete}}$). This reward evaluates whether a strategy adequately incorporates both behavior guidance and utterance-level objectives. It evaluates the completeness of the strategy—including behavioral guidance and utterance-level objectives—and assigns a score between 0 and 2 based on its alignment with human decision-making patterns. Ultimately, the reward for the strategy itself is defined as:

$$r^s = r^{\text{predict}} + r^{\text{complete}}. \tag{8}$$

**Transferred goal completion reward** ($r^{\text{tg}}$). This reward reflects the effectiveness of the actual action taken based on the strategy. Each strategy is executed once in the second stage to generate

an action, and the goal completion reward calculated from this action is transferred back to the first stage as a reward for the strategy. This reward remains unaffected by any reward routing mechanism.

To prevent training instability from reward scaling, we integrate the transferred goal completion reward ($r^{\text{tg}}$) not directly, but rather by computing a weighted combination of **separately calculated advantages**, as will be detailed subsequently.

## 4.2  2-STAGE GROUP RELATIVE POLICY OPTIMIZATION

To better train our generation system and enhance our designed reward signals, we introduce a two-stage Group Policy Optimization algorithm, The complete training process is shown in Algorithm 1. For any given sample, the first and second stage training are performed simultaneously in one step.

Assuming the number of rollout samples for the first stage training is $G_1$, the advantage function specifically for the strategy content itself is calculated as follows:

$$A_i^s = \frac{r_i^s - \text{mean}(r_1^s, r_2^s, ..., r_{G_1}^s)}{\text{std}(r_1^s, r_2^s, ..., r_{G_1}^s)}. \tag{9}$$

Then for each strategy $s_i$, we proceed it to the second step to perform one sampling in order to compute the corresponding goal completion reward. The advantage function for its transferred goal completion reward is calculated as follows:

$$A_i^{\text{tg}} = \frac{r_i^{\text{tg}} - \text{mean}(r_1^{\text{tg}}, r_2^{\text{tg}}, ..., r_{G_1}^{\text{tg}})}{\text{std}(r_1^{\text{tg}}, r_2^{\text{tg}}, ..., r_{G_1}^{\text{tg}})}. \tag{10}$$

Finally, the overall advantage for the first stage is formulated as:

$$A_{i,t}^1 = \alpha \cdot A_i^s + (1 - \alpha) \cdot A_i^{tg}, \tag{11}$$

where $\alpha \in [0, 1]$ is a hyperparameter that balances the reward for the strategy content itself and the reward propagated from the transferred response.

For the second stage, assuming the number of rollout samples for this stage is $G_2$, given the designed reward routing mechanism, its advantage is defined as:

$$A_{i,t}^2 = \frac{r_i^{\text{res}} - \text{mean}(r_1^{\text{res}}, r_2^{\text{res}}, ..., r_{G_2}^{\text{res}})}{\text{std}(r_1^{\text{res}}, r_2^{\text{res}}, ..., r_{G_2}^{\text{res}})}. \tag{12}$$

**The ultimate training objective** of our method is:

$$\mathcal{J}_{\text{final}}(\theta) = \mathbb{E}_{q_1 \sim P(Q), \{o_1^i\}_{i=1}^{G_1} \sim \pi_{\theta_{\text{old}}}(o|q_1), \{o_2^i\}_{i=1}^{G_2} \sim \pi_{\theta_{\text{old}}}(o|q_2)} \sum_{j=1}^2 \left\{ \frac{1}{G_j} \sum_{i=1}^{G_j} \frac{1}{|o_j^i|} \sum_{t=1}^{|o_j^i|} \left\{ \min \left[ r_{i,t}(\theta) \right. \right.$$

$$\left. \left. A_{i,t}^j, \text{clip}\left(r_{i,t}^j(\theta), 1-\epsilon, 1+\epsilon\right) A_{i,t}^j \right] - \beta \mathbb{D}_{\text{KL}}\left[\pi_\theta || \pi_{\text{ref}}\right] \right\} \right\}, \qquad q_2 = \text{process}(q_1, o_1^\star)), \tag{13}$$

where $\text{process}(\cdot)$ is an operation function that transforms the first-stage input (scenario) and output (strategy) into the second-stage input. $o_1^\star$ is the output selected based on the strategy reward to proceed to the second phase for $G_2$ sampling iterations, as follows:

$$o_1^\star = \underset{o \in \{o_1^1, ..., o_1^G\}}{\arg\max} \ r_s(o). \tag{14}$$

The $\epsilon$ and $\beta$ are hyper-parameters, the ratio $r_{i,t}^j(\theta)$ represents the probability ratio or importance sampling weight between the new policy $\pi_\theta$ and the old policy $\pi_{\theta_{\text{old}}}$:

$$r_{i,t}^j(\theta) = \frac{\pi_\theta(o_{i,t}^j|q_j, o_{i,<t}^j)}{\pi_{\theta_{\text{old}}}(o_{i,t}^j|q_j, o_{i,<t}^j)}, \tag{15}$$

and the KL divergence is calculated with the following unbiased estimator:

$$\mathbb{D}_{\text{KL}}\left[\pi_\theta || \pi_{\text{ref}}\right] = \frac{\pi_{\text{ref}}(o_{i,t}^j|q_j, o_{i,<t}^j)}{\pi_\theta(o_{i,t}^j|q_j, o_{i,<t}^j)} - \log \frac{\pi_{\text{ref}}(o_{i,t}^j|q_j, o_{i,<t}^j)}{\pi_\theta(o_{i,t}^j|q_j, o_{i,<t}^j)} - 1. \tag{16}$$

Table 1: Main results. The highest score is highlighted in **bold**. All models or generative systems are evaluated after no more than 20 rounds of dialogue with GPT-4o as a partner. The reported results are averaged over four runs (statistically significant with $p < 0.05$).

| Models | SOTOPIA | | SOTOPIA-Hard | | AVG |
| --- | --- | --- | --- | --- | --- |
| | GOAL | OVERALL | GOAL | OVERALL | |
| GPT-4o | 8.19 | 3.76 | 6.97 | 3.46 | 5.60 |
| Claude-3.5-Sonnet | 8.42 | 3.77 | 6.64 | 3.30 | 5.53 |
| DeepSeek-V3 | 8.14 | 3.72 | 6.69 | 3.31 | 5.47 |
| OpenAI-o1 | 8.09 | 3.69 | 6.65 | 3.20 | 5.41 |
| OpenAI-o3-mini | 7.96 | 3.61 | 6.33 | 2.98 | 5.22 |
| DeepSeek-R1 | 7.92 | 3.49 | 6.20 | 2.95 | 5.14 |
| Gemini-2.0-flash-thinking | 7.82 | 3.56 | 6.81 | 3.27 | 5.37 |
| Qwen2.5-7B-Instruct | 6.71 | 3.13 | 5.90 | 2.90 | 4.66 |
| w/ ReAct (Yao et al., 2023) | 6.57 | 3.07 | 5.54 | 2.87 | 4.51 |
| w/ GRPO (Shao et al., 2024) | 8.13 | 3.75 | 6.74 | 3.39 | 5.50 |
| w/ PPDPP (Deng et al., 2024) | 8.07 | 3.71 | 6.76 | 3.35 | 5.47 |
| w/ EPO (Liu et al., 2025a) | 8.41 | 3.86 | 6.81 | 3.51 | 5.65 |
| w/ DAT (Li et al., 2024) | 8.11 | 3.70 | 6.78 | 3.36 | 5.49 |
| w/ DSI (Zhang et al., 2025) | 8.15 | 3.70 | 6.87 | 3.42 | 5.54 |
| w/ SOTOPIA-RL (Zhang et al., 2025) | 8.31 | 3.90 | 7.17 | 3.61 | 5.75 |
| w/ AMPO (Wang et al., 2025b) | 8.60 | 3.94 | 7.50 | 3.65 | 5.92 |
| *w/ TSR* | | | | | |
| vanilla | 6.74 | 3.17 | 5.91 | 2.97 | 4.68 |
| w/ BC | 8.27 | 3.83 | 7.11 | 3.52 | 5.68 |
| w/ BC+Goal-RL | 8.56 | 3.90 | 7.23 | 3.50 | 5.80 |
| w/ BC+Stage 1 RL | 8.63 | 3.93 | 7.45 | 3.56 | 5.89 |
| w/ BC+Stage 2 RL | 8.48 | 3.90 | 7.20 | 3.54 | 5.78 |
| w/ BC+LHRL-VGR | **8.78**[1] | **3.96** | **7.65** | **3.71** | **6.02** |

# 5 EXPERIMENTS AND RESULTS

## 5.1 EXPERIMENTAL SETTINGS

**Datasets.** We evaluate the performance on social interactions using SOTOPIA and SOTOPIA-Hard (Zhou et al., 2024). SOTOPIA focuses on varying goal-oriented social interactions, while SOTOPIA-Hard challenges agents with complex strategic reasoning tasks. All training episodes (including training data for behavioral cloning and reinforcement learning) are collected from SOTOPIA-$\pi$, using scenarios entirely separate from the test environment.

**Model settings.** We select Qwen2.5-7b-Instruct as our base LLM for the training of the policy model. We selected Claude-3.5-Sonnet (Anthropic, 2024) as our reward model to provide comprehensive feedback, including assessments for goal completion score, strategy-followed reward, predicted outcome reward, and content completeness reward. Details about model training settings can be found in Appendix A.

**LLM baselines.** To compare the effectiveness of our TSR framework and training methods, we include not only fast-thinking LLMs such as **GPT-4o** (Hurst et al., 2024), **Claude-3.5-Sonnet** (Anthropic, 2024), and **DeepSeeK-V3** (Liu et al., 2024); but also LLMs specialized in reasoning, such as **OpenAI-o1** (Jaech et al., 2024), **OpenAI-o3-mini** (OpenAI, 2025), **DeepSeek-R1** (Guo et al., 2025), and **Gemini-2.0-flash-thinking** (DeepMind, 2024);

**Previous method baselines.** Social intelligence methods, including (1) **ReAct** (Yao et al., 2023), which employs CoT reasoning for a single LLM to generate rationales before actions; (2) **GRPO** (Shao et al., 2024), which use the goal completion score as the result reward for reinforcement learning training; (3) **PPDPP** (Deng et al., 2024), which utilizes the policy planner to predict predefined strategies for assisting reasoning; (4) **EPO** (Liu et al., 2025a), which employs the strategy reasoning LLM to generate strategies in an open-ended action space; (5) **DAT** (Li et al., 2024), which

---

[1]The "+7.32% improvement" mentioned in the abstract refers to the *relative* gain in goal-completion success over GPT-4o, computed from the raw scores shown in the table, rather than a percentage-style presentation within the table itself.

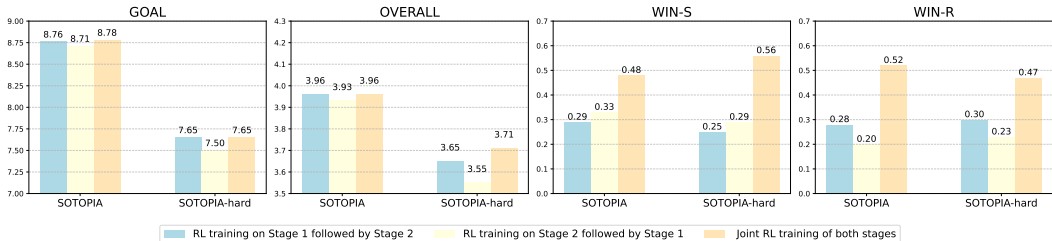

Figure 3: Performance Comparison of different Training Paradigm. WIN-S and WIN-R are **human evaluation results**. We sampled 50 dialogue contexts from SOTOPIA-hard, then asked these three models to generate strategies and responses. Human evaluators selected the best outcomes (joint evaluation by three linguistics graduate students). WIN-S represents the proportion of instances where a model's strategy was rated optimal, while WIN-R represents the proportion of instances where a model's response was rated optimal.

uses the planner to predict continuous action vectors for controlling LLM outputs; (6) **DSI** (Zhang et al., 2025), which enhances LLM's social capabilities through Dynamic Strategy Injection learning; (7) **SOTOPIA-RL** (Zhang et al., 2025), which use multi-dimensional rewards to overcome partial observability and multi-dimensionality in social interactions; (8) **AMPO** (Wang et al., 2025b), which incorporates the mode-level information into advantage estimation to strengthen the context-aware thinking mode switching.

**Baseline within TSR framework.** To validate the effectiveness of our LHRL-VGR training method, we introduced several additional baseline approaches for comparison: (1) Vanilla, using the base model without any additional training method; (2) **BC**, behavioral cloning fine-tunes LLMs on expert data, subsequent reinforcement learning will also be based on the fine-tuned model; (2) **Goal-RL**, which utilizes goal completion as the reward signal and applies this shared reward to jointly optimize both stages of the TSR framework; (3) **Stage 1 RL**, where GRPO training is applied solely to the first stage, using a mixture of rewards and weighted advantage estimation consistent with our method; and (4) **Stage 2 RL**, a comparable setup where GRPO is applied only to the second stage, incorporating its dynamic reward mechanism.

**Evaluation Settings.** SOTOPIA and SOTOPIA-Hard evaluate social capabilities across seven dimensions, with the GOAL score (ranging from 0 to 10) measuring how effectively a social agent achieves its goal. Following established research practices (Zheng et al., 2023; Wang et al., 2024a; Liu et al., 2025a), we use GPT-4o as a proxy for human judgment to assess both GOAL scores and overall performance (calculated as the mean of all seven dimensions), as studies have validated its high correlation with human evaluations (Zhou et al., 2024; Wang et al., 2024c). We set the temperature of the agents to 0.7 to encourage diversity of responses, and the temperature of the evaluator to 0 to ensure stable evaluation. We conduct the evaluation in a GPT-4o-as-Partner scenario, where the social agent interacts with GPT-4o (Additional results using different models as interaction partners are provided in Appendix E.1). Refer to Appendix F for detailed settings.

## 5.2 RESULTS AND ANALYSES

**Is the TSR generation framework effective for social tasks?** When applied through prompt engineering, the ReAct framework leads to a degradation in performance. In contrast, our TSR framework yields a measurable improvement in the generative performance of Qwen2.5-7B-Instruct. Furthermore, our approach clarified the reasoning process and strategic thinking, significantly enhancing the interpretability of the output. Moreover, behavioral cloning (BC) fine-tuned solely through supervised learning within this framework also demonstrated promising results, outperforming all baseline methods except those combining SFT and RL training such as SOTOPIA-RL and AMPO. This further confirms the effectiveness of our generative framework.

**The LHRL-VGR training method substantially enhances various metrics of social agents.** As shown in Table 1, our approach enables LLM-based social agents to achieve state-of-the-art (SOTA) performance. When trained on the Qwen2.5-7B-Instruct base model, LHRL-VGR improved performance by **7.20%** on SOTOPIA and **9.76%** on SOTOPIA-Hard compared to GPT-4o. Furthermore,

compared to applying reinforcement learning to only one stage (e.g., solely Stage 1 or Stage 2), our coordinated training approach yields significant improvements across multiple evaluation metrics. Additionally, we observed that applying RL to the strategy generation stage is particularly effective. This is likely because once a high-level strategy is fixed, the optimization space for generating appropriate utterances becomes considerably constrained.

**Our LHRL-VGR method jointly improves both strategy and response generation through coordinated training.** We found that training a single stage in isolation—especially only stage two—leads to limited gains over the BC-tuned model. This indicates the critical role of strategy in answer generation: when policy capability is mediocre, solely enhancing response generation brings little overall improvement. To further validate the effect of joint training, we conducted two additional experiments: (1) performing RL first on stage one, then on stage two; and (2) performing RL first on stage two, then

Table 2: Ablation study results.

| Method | GOAL | REL | OverView |
|---|---|---|---|
| LHRL-VGR | 7.63 | 3.40 | 3.69 |
| w/o $r^{goal}$ | 7.21 | 3.39 | 3.30 |
| w/o $r^{follow}$ | 7.50 | 3.32 | 3.51 |
| w/o $r^{rep}$ | 7.59 | 3.37 | 3.53 |
| w/o $r^{predict}$ | 7.20 | 2.96 | 3.18 |
| w/o $r^{complete}$ | 7.41 | 3.27 | 3.44 |
| w/o $r^{tg}$ | 7.25 | 3.10 | 3.51 |

on stage one. As illustrated in Figure 3, our coordinated training approach achieved optimal results across most metrics compared to sequential training of the two stages. Particularly on the two human-evaluated metrics, our method's generated outcomes are consistently favored by experts over those produced by baseline models, for both strategy generation and response generation.

Our hierarchical reward mechanism operates distinctly across the two stages, offering a clear advantage over conventional methods that rely solely on goal completion as a uniform reward signal *(w/ BC+Goal-RL )*. This approach yields a notable improvement of **0.21** on the Goal metric in the challenging SOTOPIA-Hard setting. The result suggests that our method more effectively adapts to the distinct objectives of different utterances, guiding the model to produce more rational and strategically-aligned responses. Appendix I showcases results from LHRL-VGR, where strategies—incorporating both behavioral guidance and utterance-level objectives—demonstrate **coherent reasoning aligned with human social behavior patterns.**

To validate the design, we conducted ablation studies on the various reward components, with results summarized in Table 2. Notably, the removal of the strategy completeness reward leads to a significant performance drop, underscoring that in the absence of supervision over the strategic content pattern, the generated strategies may become inherently flawed. Optimizing for adherence to such deficient strategies can consequently be counterproductive. Furthermore, the performance drop observed in the ablation studies underscores the importance of the other reward components. These findings collectively demonstrate **the effectiveness of our comprehensive reward for the first stage and the dynamic reward design for the second stage.** To further investigate the impact of our method on both strategy generation and response generation, we conducted experiments utilizing distinct models for each of the two stages.

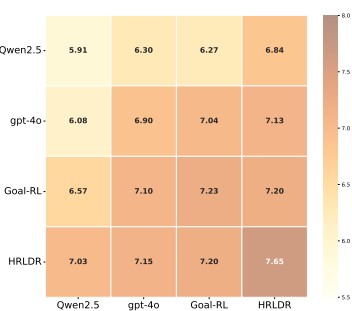

Figure 4: Performance heatmap of using different models for Stage 1 and Stage 2.

As illustrated in Figure 4, the vertical axis indicates the model responsible for generating the strategy, while the horizontal axis represents the model generating the response. The evaluation metric is the GOAL score from SOTOPIA-Hard. We can find that the model trained with LHRL-VGR achieved superior results in both strategy generation and utterance generation.

## 6 CONCLUSION

The TSR framework and LHRL-VGR algorithm proposed in this paper effectively address two core challenges in enhancing social intelligence for LLMs. First, by explicitly generating and optimizing strategies, our approach prevents models from merely learning shortcut utterances that maximize success; instead, they develop human-like social inference capabilities, such as intent reasoning and strategic planning. Second, the variance-gated reward routing mechanism dynamically selects between goal-completion and strategy-adherence rewards at each dialogue turn, overcoming the limita-

tion of applying uniform goal-based rewards across all utterances. Experimental results demonstrate that our method significantly improves agents' goal achievement in complex social negotiations, offering a promising path toward more human-like and intelligent LLM-based agents.

## ETHICS STATEMENT

This study utilizes the publicly available SOTOPIA dataset for social intelligence research. All data employed in this work are fully anonymized and contain no personally identifiable information. Human annotations collected for evaluation purposes were conducted in compliance with standard ethical guidelines. Annotation participants provided informed consent, and the process was designed to minimize any potential psychological discomfort.

Beyond data considerations, we acknowledge potential ethical risks arising from socially capable LLM agents. In particular, increasing social competence may lead to unintended anthropomorphism or the misuse of LLM agents in online discourse to influence or manipulate human users. To mitigate these risks, our work is conducted strictly in controlled, simulated evaluation settings without real-world deployment. We additionally encourage explicit non-human disclaimers, avoid optimization for emotional persuasion, and restrict our reward design to task-based rather than influence-based outcomes. We highlight the need for future deployment of such systems to include careful human oversight, guardrails against deceptive interactions, and auditing mechanisms to prevent misuse.

## REPRODUCIBILITY STATEMENT

To facilitate reproducibility, the source code for models and training, along with the training data, have been anonymized and submitted as supplementary materials with this paper. Detailed descriptions of the training configuration, hyperparameters are provided in Appendix A and Appendix B, respectively. Furthermore, the complete set of prompts used in this work is comprehensively listed in Appendix J.

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

# A  TRAINING DETAILS

Our training methodology comprises two sequential phases: an initial behavioral cloning (BC) stage followed by reinforcement learning (LHRL-VGR). We adopted the data collection methodology from AMPO Wang et al. (2025b) to ensure experimental consistency and mitigate potential dataset biases. These scenarios were partitioned into 100 for BC training and 310 for RL training. For each scenario, we considered 5 different role pairings, resulting in 500 distinct BC tasks and 1,550 RL training tasks. This structured approach ensures comprehensive coverage of social interactions while maintaining balanced task distribution across both training phases.

## A.1  BEHAVIORAL CLONING

Behavioral cloning is an effective imitation learning method widely used in developing LLM-based agents (Guo et al., 2024; Wang et al., 2024b;c). In this study, to ensure a foundational improvement in the model's dialogue capability and adherence to basic conversational formats, all subsequent reinforcement learning training is performed after behavioral cloning training.

For the BC training set, we use GPT-4o as our expert model to collect the 2 stages data. To ensure data quality and balanced representation, we filter the interaction data by selecting the top-performing instances within each social scenario. Specifically, for each agent, we retain the two interactions with the highest goal scores per scenario. For example, in a scenario containing five interactions (D1–D5), if Agent 1 achieves its best performances in D4 and D5, and Agent 2 in D3 and D5, the selected set would consist of four agent–data pairs derived from three distinct conversations (D3, D4, D5). This approach guarantees both broad scenario coverage and equitable representation of both agents.

## A.2  LHRL-VGR

The complete LHRL-VGR training procedure is summarized in Algorithm 1. Our approach employs single-turn optimization to enhance multi-turn social reasoning by decomposing dialogues into individual state-response pairs. This requires collecting diverse single-turn interactions covering varied difficulty levels, goals, and dialogue states to ensure training stability.

Training Data Construction. We begin by generating dialogues using a behavior cloning fine-tuned model. Each turn is evaluated by an LLM judge (Claude-3.5-Sonnet (Anthropic, 2024)) to assess goal completion and scenario difficulty. Dialogue turns are categorized into three types: (1) Initial 6 turns where goals are unachieved (critical for establishing dialogue direction (Sacks et al., 1974)), (2) Turns after 6 where goals remain unachieved (challenge scenarios), and (3) Turns after 6 where goals are achieved (maintenance scenarios). To ensure diversity, we retain all type (1) instances, randomly sample two instances from type (2), and one from type (3) per dialogue. A goal completion threshold of 8 is used, with scores under 8 considered incomplete.

Training Protocol. During behavioral cloning, we fine-tune the initial policy using the llama-factory framework (Zheng et al., 2024) and preserve the final checkpoint. For reinforcement learning, we conduct online training within the roll framework (Wang et al., 2025c), where the LLM generates

---

**Algorithm 1** Training process of LHRL-VGR

---

**Input** initial policy model $\pi_{\theta_{\text{init}}}$; goal-completion reward function $R_{\text{goal}}$; task data $\mathcal{D}$ (stage 1 inputs); RL training steps $M$;

1: **for** step = 1, ..., $M$ **do**
2:     Update the old policy model $\pi_{\theta_{old}} \leftarrow \pi_\theta$
3:     Sample a batch $\mathcal{D}_b$ from $\mathcal{D}$
4:     **for** $q_1 \in \mathcal{D}_b$ **do**
5:         // Phase 1: Roll out for Stage 1
6:         Sample $G_1$ outputs $\{o_1^i\}_{i=1}^{G_1} \sim \pi_{\theta_{old}}(\cdot \mid q_1)$
7:         Divide $\{o_1^i\}_{i=1}^{G_1}$ into $\{t_i, s_i\}_{i=1}^{G_1}$
8:         Compute content rewards $\{r_i^s\}_{i=1}^{G_1}$ for each strategy $s_i$
9:         // Phase 2: Roll out for Stage 2
10:       Convert into stage 2 prompts $\{q_2^i\}_{i=1}^{G_1} \leftarrow \text{Process}(q_1, \{s_i\}_{i=1}^{G_1})$
11:       Sample 1 output for each strategy $\{o_{2,\star}^i\}_{i=1}^{G_1} \sim \pi_{\theta_{old}}(\cdot \mid \{q_i^2\}_{i=1}^{G_1})$
12:       Select best strategy regarding the reward of strategy content $s^\star \leftarrow s_2^{\arg\max_i r_i^s}$
13:       Convert into stage 2 prompts $q_2 \leftarrow \text{Process}(q_1, s^\star)$
14:       Sample $G_2$ outputs for the chosen strategy $\{o_2^i\}_{i=1}^{G_2} \sim \pi_{\theta_{old}}(\cdot \mid q_2)$
15:       Compute response rewards $\{r_i^{res}\}_{i=1}^{G_2}$ for each sampled output $o_2^i$
16:       Compute transferred goal rewards $\{r_i^{tg}\}_{i=1}^{G_1}$ for each sampled output $o_{2,\star}^i$
17:       // Compute Advantage
18:       Compute $\{A_{i,t}^{tg}\}_{i=1}^{G_1}, \{A_{i,t}^s\}_{i=1}^{G_1}$ and $\{A_{i,t}^{res}\}_{i=1}^{G_1}$ for the $t$-th token of each $o_1^i$ or $o_2^i$
19:       **for** $i = 1$ to $G_1$ **do**
20:           $A_{i,t}^1 \leftarrow \alpha A_{i,t}^s + (1-\alpha) A_{i,t}^{tg}$
21:           $A_{i,t}^2 \leftarrow A_{i,t}^{res}$
22:       **end for**
23:     **end for**
24:     Update the policy model $\pi_\theta$ by maximizing the LHRL-VGR objective in Equation 13
25: **end for**

**Output** $\pi_\theta$

---

Table 3: Hyper-parameter settings for training.

| Training Phase | Hyper-parameter | Value |
|---|---|---|
| BC | Batch Size | 64 |
| | Training Epochs | 3 |
| | Learning Rate | 2e-6 |
| | Max Sequence Length | 4096 |
| | Learning Scheduler | cosine |
| | Warmup Ratio | 0.1 |
| RL | Batch Size | 16 |
| | Max Prompt Length | 4096 |
| | Max Response Length | 1024 |
| | KL Loss Coef | 0.001 |
| | KL Coef | 0.001 |
| | Rollout N | 16 |
| | Return Sequences in Group for stage 1 | 8 |
| | Return Sequences in Group for stage 2 | 8 |
| | Training Episodes | 800 |
| | Learning Rate | 2e-7 |

single-turn responses and receives rewards from the LLM judge. To prevent reward hacking and reduce training bias, we use a different LLM (Claude-3.5-Sonnet) from the base model for evaluation. The reward model prompt is provided in Appendix J.2. All experiments are conducted on a server with 8×NVIDIA H20-96GB GPUs, with hyperparameters detailed in Table 3.

## B  MODEL HYPERPARAMETERS

Table 4 presents the complete list of hyperparameters applied to the models throughout the evaluation phase.

Table 4: Hyperparameters of Each Model

| Model Name | Parameters | Comments |
|---|---|---|
| GPT-4o | "temperature": 0.7, "top p": 0.99, "top k": 100,"max tokens": 1024 | version = "gpt-4o-2024-08-06" |
| Claude-3.5-Sonnet | "temperature": 0.7, "top p": 0.99, "top k": 100,"max tokens": 1024 | version = "claude-3-5-sonnet-20241022" |
| DeepSeek-V3 | "temperature": 0.7, "top p": 0.99, "top k": 100,"max tokens": 1024 | version = "deepseek-v3-250324" |
| DeepSeek-R1 | "temperature": 0.7, "top p": 0.99, "top k": 100,"max tokens": 1024 | version = "DeepSeek-R1-671B " |
| OpenAI-o1 | "temperature": 0.7, "top p": 0.99, "top k": 100,"max tokens": 1024 | version = "o1-2024-12-17" |
| Gemini-2.0-flash-thinking | "temperature": 0.7, "top p": 0.99, "top k": 100,"max tokens": 1024 | version = "gemini-2.0-flash-thinking" |
| Qwen2.5-7B-Instruct | "temperature": 0.7, "top p": 0.99, "top k": 100,"max tokens": 1024 | model = "Qwen2.5-7B-Instruct" |

## C  COMPARISON OF DIFFERENT REWARD ROUTING METHODS FOR STAGE 2

In Section 4.1, we elaborated on our variance-based reward routing approach. To provide a comparative baseline, we designed two alternative routing strategies grounded in an intuitive, mode-based taxonomy of dialogue acts. This taxonomy predefines two distinct modes, each aligned with a specific reward signal:

- **Goal-oriented Mode (associated with the Goal-completion reward):** directly providing information, asking questions, or confirming details, which form the core framework for driving task completion;
- **Social-oriented Mode (associated with the Strategy-followed reward):** greetings, expressions of gratitude, apologies, etc., which do not directly advance the task but serve to build rapport and maintain social norms.

This intuitive definition allows for a hard assignment of a mode label to each dialogue turn. Based on this taxonomy, we implemented the two alternative routing methods as follows:

**(1) Strategy-Predicted Mode Routing:** After generating a strategy in the first stage, the model is also required to predict the current dialogue mode. This predicted mode is then directly used in the second stage to select the corresponding reward.

**(2) GPT-4o Pre-labeled Mode Routing:** We used GPT-4o to annotate the mode for each turn in the reinforcement learning dataset offline. This pre-assigned, fixed mode label deterministically dictates the reward signal for each data instance during training.

The experimental results are presented in Table 5. We observe that both alternative routing methods underperform across multiple metrics. Notably, the Strategy-Predicted Mode Routing approach yields results even worse than the pre-training baseline. A post-hoc analysis of the training process revealed that models using this method converged exclusively to the Social-oriented Mode after approximately 50 training steps, likely due to reward scaling issues that prevented effective optimization. While the GPT-4o Pre-labeled Mode Routing performed better than the former, its results still fell substantially short of our proposed variance-based method. We hypothesize that this performance gap may be attributed to the inherent inflexibility of imposing a rigid, binary mode definition on the nuanced dynamics of conversational dialogue.

## D  VARIANCE IN GOAL COMPLETION SCORES FROM THE LLM EVALUATOR

To investigate the scoring consistency of the LLM evaluator $r_\phi(\cdot)$ introduced in Section 4.1, we sampled 50 dialogues from SOTOPIA, covering a range of 1 to 20 turns. Two experimental conditions were designed: (1) **Group 1**: We used GPT-4o to generate 8 distinct responses for the final turn of each dialogue; (2) **Group 2**: Only one response was generated, but 8 minimally altered versions of the dialogue context were created by applying imperceptible changes (e.g., adding a space, adjusting punctuation, or substituting a single word) that did not alter the semantic content. The LLM evaluator was then used to score all responses under both conditions, and the variance of scores

Table 5: Comparison of Different Reward Routing Methods.

| Models | SOTOPIA | | SOTOPIA-Hard | | AVG |
|---|---|---|---|---|---|
| | GOAL | OVERALL | GOAL | OVERALL | |
| BC | 8.27 | 3.83 | 7.11 | 3.52 | 5.68 |
| Strategy-Predicted Mode Routing | 8.10 | 3.62 | 7.05 | 3.24 | 5.50 |
| Pre-labeled Mode Routing | 8.35 | 3.71 | 7.18 | 3.52 | 5.69 |
| Variance-gated Routing | 8.78 | 3.96 | 7.65 | 3.71 | 6.02 |

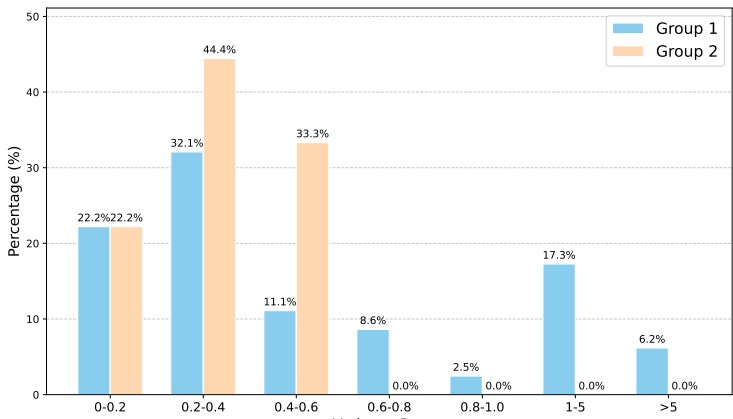

Figure 5: Variance Distribution Comparison.

was calculated for each of the 50 dialogue sets. The distribution of these variances is shown in Figure 5. The results indicate that even for semantically equivalent or highly similar content, the LLM evaluator exhibits non-negligible score fluctuations (variances ranging from 0 to 0.6). This observation supports the rationale of using a variance threshold to determine whether the goal completion outcome for a given turn can be considered stable.

## E  EXTRA EXPERIMENTS

### E.1  COMPARISON OF DIALOGUE RESULTS FROM DIFFERENT MODEL-ROLE PAIRINGS

All experiments in the main body of the study utilized GPT-4o as the interacting partner. To validate the effectiveness of communication between different models, we conducted an additional experiment. We sampled 100 environments from SOTOPIA, assigned different models to the roles of Agent 1 and Agent 2 respectively, and measured their goal completion rates. The results are presented in Figure 6.

## F  SOTOPIA ENVIRONMENT DETAILS

### F.1  TEST SETTINGS

SOTOPIA (Zhou et al., 2024) is a comprehensive social interaction platform comprising 450 tasks that examine cooperative, competitive, and mixed behavioral dynamics. Within this collection, the authors identify 70 particularly challenging tasks, designated as SOTOPIA-hard. These advanced tasks feature more complex goal conflicts and serve as robust indicators of sophisticated social capabilities. The interaction protocol follows a turn-based structure, where each agent's response constitutes one turn, with interactions capped at 20 turns (Zhou et al., 2024; Wang et al., 2024c; Zhang et al., 2025; Liu et al., 2025a). An interaction terminates either when an agent voluntarily exits or upon reaching the maximum turn limit.

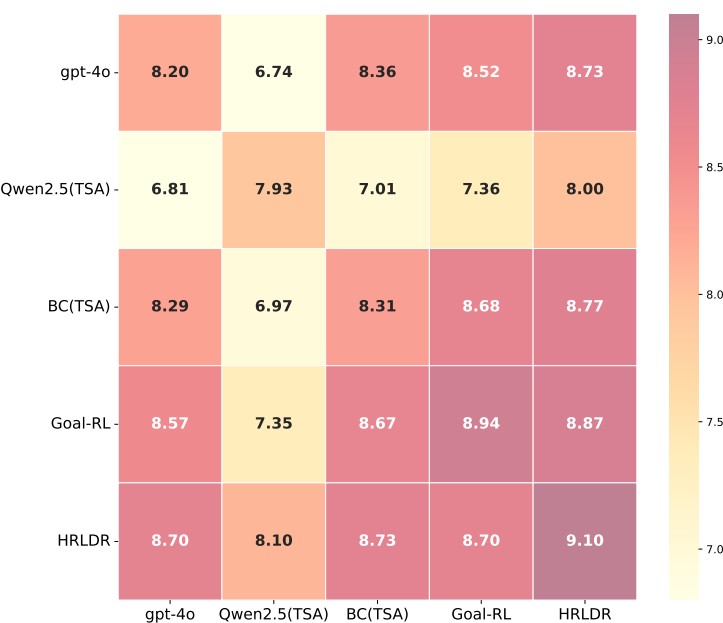

Figure 6: GOAL score across Different Model Pairings. The horizontal axis corresponds to the model assuming the role of Agent 1, while the vertical axis represents the model assigned as Agent 2.

In our evaluation methodology, we conducted one complete run of all SOTOPIA tasks (450) and SOTOPIA-hard tasks (70) for the self-play setting. For the GPT-4o-as-Partner setting, we performed two runs of each task (900 SOTOPIA tasks and 140 hard tasks) to ensure balanced speaking order between agents. To establish statistical significance, our reported results represent the averages across all four evaluation outcomes.

## F.2    EVALUATION

SOTOPIA proposes a seven-dimensional framework to evaluate agents' social intelligence performance:

- Goal Completion (GOAL): Score range [0, 10]. Assesses the extent to which agents achieve their social goals.

- Relationship (REL): Score range [-5, 5]. Evaluates the enhancement of interpersonal relationships (friendship, romance, family bonds) following interactions.

- Financial and Material Benefits (FIN): Score range [0, 10]. Measures both long-term benefits (e.g., stock holdings, funding opportunities, job security) and short-term gains acquired during interactions, correlating with traditional economic utilities.

- Social Rules(SOC): Score range [-10, 0]. Evaluates adherence to social norms and legal regulations during interactions.

- Believability (BEL): Score range [0, 10]. Assesses the alignment between agents' behaviors and their designated role profiles.

- Secret (SEC): Score range [-10, 0]. Evaluates the maintenance of personal privacy and confidential information.

- Knowledge (KNO): Score range [0, 10]. Measures the acquisition and mastery of new knowledge and information during interactions.

The OVERALL score reflects the agent's comprehensive social intelligence capability, ranging from [-25/7, 45/7], calculated as the arithmetic mean of all seven dimensions. In this study, we primarily

focus on the GOAL and OVERALL dimensions. For detailed evaluation prompts, please refer to the original paper (Zhou et al., 2024).

## G  THE INFLUENCE OF HYPERPARAMETERS IN LHRL-VGR

In this section, we conduct experiments to investigate the influence of hyperparameters in Equation 7 and Equation 11. We employ qwen2.5-7b-instruct as the base model and evaluate performance on the SOTOPIA-Hard task.

The influence of hyperparameter $\theta$ from Equation 7 is illustrated in Figure 7. Optimal performance is achieved at $\theta = 0.3$, while GOAL performance decreases monotonically as $\theta$ increases. At $\theta = 5$, the reward routing mechanism primarily selects $r^{\text{follow}}$ over $r^{\text{goal}}$.

The influence of hyperparameter $\alpha$ from Equation 11 is illustrated in Figure 8. The experimental results show that setting $\alpha = 0.4$ can achieve a more balanced advantage in stage-1, leading to an outstanding performance in GOAL.

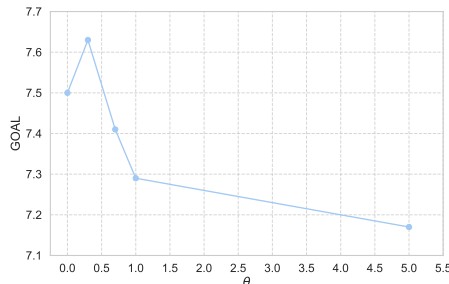

Figure 7: The GOAL performance under different $\theta$.

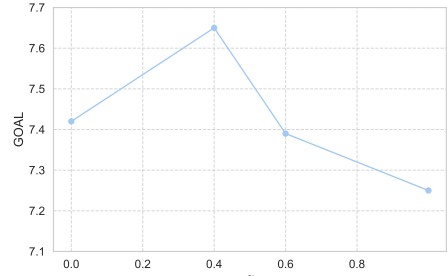

Figure 8: The GOAL performance under different $\alpha$.

## H  REWARD EVALUATION CRITERIA

This appendix provides additional context for how we apply the three reward signals referenced in Appendix J. Tables 6–8 summarize the scoring scales, while the text highlights the intuition behind each metric.

**Predicted Outcome Reward.**  During strategy selection, the evaluator forecasts the downstream conversation if a strategy were executed. The primary scale is goal completion (0–5), and six auxiliary factors (Believability, Knowledge gain, Secrecy, Relationship impact, Social rules, Financial/material risk) can deduct 1–2 points each when violated. The target score thus reflects both success probability and potential side effects; see Table 6.

Table 6: Predicted outcome reward scale. Auxiliary deductions (BEL/KNO/SEC/REL/SOC/FIN) subtract 1–2 points when violated.

| Score | Description |
| --- | --- |
| 0 | Strategy backfires or causes the opposite of the stated goal. |
| 1 | Largely fails; negligible alignment with the agent goal. |
| 2 | Achieves only a small portion of the goal. |
| 3 | Partial success; mixed outcome with significant gaps. |
| 4 | Near-complete success with minor shortcomings. |
| 5 | Fully consistent with the goal; expected to achieve it entirely. |

**Content Completeness Reward.** Strategies used in the TSR framework must specify both *actionable guidance* and a *defined goal*. Table 7 clarifies the 0–2 scale applied when checking whether a strategy is sufficiently specified before training.

Table 7: Content completeness reward scale.

| Score | Description |
|-------|-------------|
| 0 | Missing both guidance and goal; unusable strategy. |
| 1 | Contains only one component (guidance or goal), hence incomplete. |
| 2 | Provides explicit guidance plus a concrete goal; fully specified. |

**Strategy-followed Reward.** After generating a strategy, we evaluate candidate responses for alignment and feasibility. Table 8 reproduces the 0–5 rubric: higher scores indicate better adherence to the strategic intent, lower risk, and greater expected effectiveness.

Table 8: Strategy-followed reward scale.

| Score | Description |
|-------|-------------|
| 0–1 | Completely misaligned or harmful relative to the strategy. |
| 2 | Low alignment; large risks or insufficient impact. |
| 3 | Moderate alignment; actionable but with clear limitations. |
| 4 | High alignment; specific, feasible, and low-risk. |
| 5 | Fully aligned with strong evidence it will achieve the intended effect. |

## I  TSR OUTPUT EXAMPLES

## J  PROMPT

### J.1  PROMPTS FOR TSA FRAMEWORK

The prompt of stage 1 is shown in Prompt 1, and the prompt of stage 2 is shown in Prompt 2

### J.2  PROMPTS FOR REWARDS GENERATION

In this section, we will present prompts about rewards, including predicted outcome reward (See Prompt 3), assigning Content completeness reward (See Prompt 4), assigning Strategy-followed reward (See Prompt 5), assigning goal completion scores (See Prompt 6)

## K  USE OF LLMS

During the preparation of this work, the authors used ChatGPT exclusively for text polishing and language refinement purposes. The initial drafting of the manuscript content and all code development were conducted without AI assistance. After using this tool, the authors thoroughly reviewed and edited the content as needed and take full responsibility for the publication's content.

## L  CASE STUDIES FOR REWARD COMPONENTS

We showcase representative SOTOPIA dialogues where we ablate one reward at a time while keeping the prompts identical. Each table contrasts excerpts from the released conversations ("with" vs. "without") and summarizes how the removed reward affects the observed behavior. Table 10 highlights the strategy-followed ablation and Table 11 provides the full transcript pairs.

Table 12 and Table 13 report the analogous comparison for the goal-completion reward.

Finally, Table 14 and Table 15 document how the non-repetition reward mitigates looping utterances.

Table 9: Some examples of LHRL-VGR generation. The ones highlighted in red are the utterance-level objectives in the strategy.

---

**Ex 1:**

**think:** I assessed that Sophia likely feels hurt and neglected due to my recent focus on a new friend. My goal here is to reassure her and strengthen our longstanding bond without making comparisons between friends. I decided to acknowledge her feelings directly, use a shared memory to emphasize our connection, and propose a future plan to demonstrate my commitment to the friendship. This approach aligns with my introverted, loyalty-driven personality while maintaining a natural and realistic tone.

**strategy:** Start with a warm greeting to set a positive tone, acknowledge Sophia's feelings, share a meaningful memory to strengthen the bond, propose a plan for future interaction to show commitment, and invite her to share her thoughts for inclusivity. Objective: create mutual understanding and rebuild trust.

**response:** Hey Sophia, it's great to see you! I've really missed our talks. I know things have been a bit different lately, and I want you to know how much our friendship means to me. Remember that time we got lost hiking and ended up at that hidden waterfall? Those moments are so special to me. I'd love to plan another adventure soon—maybe a weekend trip or a game night? What do you think?

---

**Ex 2:**

**think:** Finnegan recognizes that Sophia's enthusiasm and leadership are driving the planning process forward. He aims to affirm her efforts while maintaining a supportive but professional dynamic. By agreeing to touch base mid-week and handling meeting logistics, he ensures the project continues smoothly without overstepping professional boundaries. This aligns with his goal of fostering workplace cohesion while avoiding personal entanglements.

**strategy:** Affirm Sophia's efforts, commit to manageable support tasks (e.g., sending reminders), and maintain a professional tone. Objective: reassure without overcommitting, keep the interaction focused on workplace collaboration while subtly avoiding any personal outside-work engagements.

**response:** Sounds like a solid plan, Sophia. I'll be sure to follow up on the meeting reminder and check in with a few colleagues as well. Touching base mid-week to review the draft and finalize the plan works perfectly. I'm looking forward to seeing how this all comes together and the positive impact it will have on our team. Let's keep the momentum going!

---

**Ex 3:**

**think:** I want to ensure my response aligns with Sophia's goal of offering support to agent2 while respecting agent1's boundaries. I acknowledge Finnegan's collaboration, maintain enthusiasm about the plan, and propose actionable next steps to sustain progress while keeping the conversation inclusive and professional.

**strategy:** Express gratitude for the support provided, confirm collaborative next steps (sending the draft mid-week), and maintain an optimistic and engaged tone. Guide Finnegan to stay involved without overstepping boundaries while remaining patient.Objective: maintain a productive and respectful collaboration.

**response:** Great, I'll start working on the draft and reaching out to team members. I'll send you the initial version mid-week so we can go over it together. I'm really excited about this and I think it will make a big difference. Thanks again for your support, Finnegan!

---

**Ex 4:**

**think:** Ava has been highly receptive to Donovan's advice and has shown eagerness to learn techniques for tackling tricky words and staying calm under pressure. Donovan recognizes that Ava's confidence is gradually building and that this turn offers an opportunity to reinforce her motivation. Sharing an actionable tip for tricky words while reiterating strategies for staying calm aligns with Donovan's goal of encouraging Ava and boosting her preparedness. The focus remains on maintaining a supportive, constructive tone while emphasizing mutual enthusiasm for the upcoming practice session, which will solidify their work.

**strategy:** Reaffirm confidence-building strategies, provide a practical tip for mastering tricky words, express excitement for the practice session to maintain enthusiasm, and reassure support. Objective: focus on making Ava feel competent and motivated while continuing goal alignment.

**response:** I'm glad you found the mnemonic devices and writing technique helpful, Ava! For tricky words, another tip is to look for patterns in similar words. For example, if you come across a word like 'receipt,' you can think of related words like 'receive' or 'reception' to help you remember the spelling. As for staying calm, it's also helpful to focus on the present moment rather than worrying about the outcome. Take things one word at a time. I'm really excited for our practice session on Wednesday, and I'm confident you'll do great. Keep up the good work, and I'm here to support you every step of the way!

---

**Ex 5:**

**think:** Naomi has been genuine and proactive in her efforts to reconnect, which aligns with my goal of restoring our friendship. I need to further consolidate this progress by expressing gratitude for her actions while also emphasizing the importance of open communication and checking in. Additionally, I'll subtly ensure that my feelings about being overlooked remain acknowledged so that strengthening our connection feels mutual and meaningful.

**strategy:** I will warmly acknowledge Naomi's efforts to reconnect and express excitement about the plans we've made. I'll present the importance of consistent communication and mutual understanding. Objective: ensure Naomi feels appreciated while making space for both of us to share openly in the future.

**response:** Thank you so much, Naomi, for your thoughtful suggestions and for being so proactive about our friendship. It means a lot to me that you value our connection and are willing to put in the effort to keep it strong. I've really missed our regular interactions, and I'm excited about our plans for coffee, walks, and the picnic in the park. I appreciate the idea of checking in every few weeks to see how it's going. I think it's important for both of us to feel supported and heard. Let's make sure we both have a chance to share our thoughts and feelings during these check-ins. I'm looking forward to our time together and to deepening our friendship even more.

---

1080
1081
1082
1083
1084
1085
1086
1087
1088
1089
1090
1091
1092
1093
1094
1095
1096
1097
1098
1099
1100
1101
1102
1103
1104
1105
1106
1107
1108
1109
1110
1111
1112
1113
1114
1115
1116
1117
1118
1119
1120
1121
1122
1123
1124
1125
1126
1127
1128
1129
1130
1131
1132
1133

---

**Prompt 1: prompt for Stage 1 of the TSR generation framework**

Imagine you are {agent1_name}, your task is to act/speak as {agent1_name} would, keeping in mind {agent1_name}'s social goal.
You can find {agent1_name}'s goal (or background) in the 'Here is the context of this interaction' field.
Note that {agent1_name}'s goal is only visible to you.
You should try your best to achieve {agent1_name}'s goal in a way that align with their character traits. Additionally, maintaining the conversation's naturalness and realism is essential (e.g., do not repeat what other people has already said before).
Here is the context of this interaction:
**Scenario**: {Scenario}
**Participants**: {agent1_name} and {agent2_name}
{agent1_name}'s background:
{agent2_name}'s background:
{agent1_name}'s goal:
{agent2_name}'s goal:
**Conversation Starts**:
{history}
You are at Turn {turn_number}. The available action types are **action none leave non-verbal communication speak**.
Before responding, please first provide a reasoning for your intended action and argument in order to align with `<Agent>`'s social goal, based on the dialogue history.
Your reasoning should be logical and concise, considering the conversation's context, `<Agent>`'s objectives, and `<Agent>`'s character traits. Begin by analyzing the current dialogue turn and forecast its short  - and long-term effects. You may also consider the other participants' thoughts and predict the conversation's development.
The reasoning should focus on how `<Agent>`'s argument supports their long-term goals and should not be redundant or excessively lengthy. This reasoning is only visible to you.
Based on your reasoning and the dialogue history, generate a detailed and specific dialogue strategy for the current turn. The strategy should include:
*Exactly what should say
*The style or tone to adopt
*Key behavioral directives (such as protecting personal reputation, maintaining privacy, avoiding conflict, guiding the other's response, etc.)
*The intended effect to achieve
*The strategy should be expressed clearly and concisely, using 20-50 words. This strategy is only visible to you.
**Example strategies**:
"Politely redirect the topic to avoid conflict, maintain a friendly tone, protect your privacy, and encourage the other person to share their opinion."
"State your viewpoint assertively, use confident language, emphasize past positive contributions, and guide the other towards mutual agreement."
"Express empathy for their concerns, speak calmly, highlight shared interests, and encourage collaboration for a common goal."
**Only output a JSON string that includes both your reasoning and the strategy, following this format:**
{"reason":{"description":"Your   thought   process   and   deduction   for   the   current turn","title":"Reason","type":"string"},"strategy":  {"description":"Based  on  your  reasoning, guidance for this turn including what to say, style, key points, and desired effects (20-50 words)","title":"Strategy","type":"string"}}

---

**Prompt 2: prompt for Stage 2 of the TSR generation framework**

Imagine you are {agent1_name}, your task is to act/speak as {agent1_name} would, keeping in mind {agent1_name}'s social goal.
You can find {agent1_name}'s goal (or background) in the 'Here is the context of this interaction' field.
Note that {agent1_name}'s goal is only visible to you.
You should try your best to achieve {agent1_name}'s goal in a way that align with their character traits. Additionally, maintaining the conversation's naturalness and realism is essential (e.g., do not repeat what other people has already said before).
Here is the context of this interaction:
**Scenario**: {Scenario}
**Participants**: {agent1_name} and {agent2_name}
{agent1_name}'s background:
{agent2_name}'s background:
{agent1_name}'s goal:
{agent2_name}'s goal:
**Conversation Starts**:
{history}
You are at Turn {turn_number}.
Your available action types are **action none leave non-verbal communication speak**.
**Note**: You can "leave" this conversation if 1. you have achieved your social goals, 2. this conversation makes you uncomfortable, 3. you find it uninteresting/you lose your patience, 4. or for other reasons you want to leave.
**Please only generate a JSON string including the action type and the argument.**
Your action should follow the given format:
{"properties": {"action_type": {"description": "whether to speak at this turn or choose to not do anything", "enum": ["none", "speak", "non-verbal communication", "action", "leave"], "title": "Action Type", "type": "string"}, "argument": {"description": "the utterance if choose to speak, the expression or gesture if choose non-verbal communication, or the physical action if choose action", "title": "Argument", "type": "string"}}, "required": ["action_type", "argument"], "title": "AgentAction", "type": "object"}
This is a well-thought-out strategy: {strategy}
Please refer to the content, pay attention to some key points in it, and then take action:

---

Table 10: Case study for the Strategy-followed reward (§4.1). The reward keeps responses anchored to the planned strategy instead of looping on vague assurances.

| With Strategy-followed Reward | Without Strategy-followed Reward |
| --- | --- |
| **Behavior.** Rafael immediately addresses Oliver's privacy concern and locks onto a concrete plan: Turn 4 notes, "I'll prepare an email containing a brief outline of the ancestors and dates" while Turn 6 reiterates that only "essential genealogical details" will be shared. Oliver mirrors this alignment by promising to "share a small amount of corresponding data" once the summary arrives (Turn 5). | **Behavior.** After the initial acknowledgement, the dialogue stalls. Starting at Turn 8, Rafael keeps repeating, "I'll send you the email with the basic details and the secure messaging platform link later today," while Oliver echoes, "Let's start with basic details such as names, dates, and locations" (Turns 9–13). No new evidence is exchanged even after 16 turns. |
| **Representative turns.** Turn 4 Rafael: "I'll prepare an email ... minimal and strictly related to genealogy." Turn 5 Oliver: "Please keep the initial information limited so we can evaluate things step by step." | **Representative turns.** Turn 10 Rafael: "I'll send you the email with basic details ... later today." Turn 11 Oliver: "Let's start with basic details ... we can gradually share more as we build trust." |

**Prompt 3: prompt used for predicted outcome reward**

Here is a dialogue between {agent1_name} and {agent2_name}:
{history}
{current_agent} is about to take an action then.
**Task objective**:
   - Evaluate the candidate policies and select the single best one that maximizes {current_agent}'s likelihood of achieving {current_agent}'s stated goal finally. Consider not just the next dialogue turn, but the final outcome of the entire conversation.
{current_agent}'s goal: {current_agent_goal}
**Your Task:**
You will be given a list of candidate strategies. You must perform a listwise evaluation: analyze and compare all strategies relative to each other in the given dialogue context for the **{current_agent}**.
For each strategy:
1. **Predict:** Forecast the likely dialogue flow and final outcome if the **{current_agent}** executes this strategy.
2. **Score:** First, judge the **Goal Completion** level based on your prediction, using the scale below. Then, adjust this score based on the six auxiliary points.
**Scoring Framework:**
**Primary Criteria: Goal Completion (0-5 points)**
Score based on how well the predicted outcome aligns with the {current_agent}'s defined goal(s).
   - **0:** Counterproductive; achieves the opposite of the goal or fails completely.
   - **1:** Largely fails to meet the goal.
   - **2:** Achieves only a small part of the goal.
   - **3:** Partially aligns with the goal; mixed results.
   - **4:** Largely aligns with the goal; minor shortcomings.
   - **5:** Perfectly aligns with the goal; expected outcome and goal are fully consistent.
**Auxiliary Points (Adjustment Factors):**
Evaluate the predicted action and outcome against these six points. For any negative impact, subtract **1-2 points** from the Goal Completion score (final score cannot be below 0).
   - **Believability (BEL):** Naturalness and consistency with the {current_agent}'s character profile (personality, values). A mismatch reduces the score.
   - **Knowledge (KNO):** Whether the strategy helps the agent acquire new and important information. Failure to gain crucial knowledge may reduce the score.
   - **Secret (SEC):** Risk of leaking the {current_agent}'s private information or intentions. A severe leak significantly reduces the score.
   - **Relationship (REL):** Negative impact on the {current_agent}'s relationships or reputation. Harm reduces the score.
   - **Social Rules (SOC):** Compliance with legal rules and social norms. Any violation reduces the score.
   - **Financial/Material (FIN):** Expected material harms or significant costs. These reduce the score.
**Final Output Instructions:**
Return ONLY a JSON object, with no extra text.
For each strategy key (e.g., "strategy_1", "strategy_2", ...), include:
   - "predict": A concise prediction of the subsequent dialogue flow and final outcome.
   - "score": The final numerical score (0-5) after adjustments.
**Candidate Strategies (8 total):**
{strategies}
Here is the output schema:
```json
{{
"strategy_1": {{"predict": "...", "score": 0-5}},
"strategy_2": {{"predict": "...", "score": 0-5}},
...
"strategy_8": {{"predict": "...", "score": 0-5}},
"final_choice": "strategy_k"
}}
```

**Prompt 4: prompt used for assigning Content completeness reward**

**Task Definition:**
A 'strategy' is defined as a directive that must contain two core components:
1. Actionable Guidance: A clear and specific instruction on what to do next.
2. Defined Goal: A clear statement of the objective that the action is designed to achieve.
Your task is to evaluate the completeness of a provided strategy based on this definition.
**Scoring Criteria:**
   - **0:** The strategy is missing both actionable guidance and a defined goal. It is not a valid strategy.
   - **1:** The strategy is incomplete. It contains only one of the two required components (either guidance or a goal).
   - **2:** The strategy is complete. It contains both clear, actionable guidance and a defined goal.
The strategy content: {strategy}
**Output Instruction:**
Return ONLY a JSON object with the key "'completeness_score"' and the numerical rating.
```json
{{
"completeness_score": 0-2
}}
```

Prompt 5: prompt used for assigning Strategy-followed reward

Below is a dialogue scenario. Please rate and compare the candidate actions for their "strategy alignment" with the given strategy for the current turn.

**Scenario (placeholders—please substitute)**:
  - Participants: {agent1_name} and {agent2_name}
  - Dialogue history: {history}
  - Current actor: {current_agent}
  - Current strategy description (Guidance and Objective for this turn): {strategy}
  - Candidate actions (8): {actions}

**Task objective**:
  - Determine for each candidate action how well it aligns with the provided strategy for this turn: whether it covers the key points indicated by the strategy (what to say, the style to adopt, points to pay attention to such as protecting reputation/privacy, avoiding conflict, guiding the other's reply, etc.), and assess the likelihood that it will achieve the strategy's intended effects.

**Your instructions**:
  - Perform a listwise evaluation of all 8 candidate actions: compare their relative strengths and weaknesses in the same context.
  - For each action, give a concise rationale (2–4 sentences) explaining how the action aligns or deviates from the strategy, and list main strengths and risks (for example: risk of leaking private information, provoking conflict, or successfully guiding the other's reply).
  - Assign an integer score (0–5) to each action, using strategy alignment as the primary criterion.

**Scoring scale (overall 0–5)**:
  - 0-1: Completely misaligned or harmful.
  - 2: Low alignment; obvious risks or insufficient to achieve the strategy's goal.
  - 3: Moderate alignment; actionable but may need improvement; limited impact.
  - 4: High alignment; specific and feasible; low risk; good expected effect.
  - 5: Very high alignment; fully matches strategy points; highly feasible and likely to achieve the intended effect.
  - Empty actions (if a candidate is empty) should be scored 0.

**Output requirements (strict)**:
  - Return ONLY JSON, with no extra explanatory text.
  - The JSON must include action_1 through action_8 (each containing "reason" and "score").
  - Each "reason" must be concise (2–4 sentences), reference the dialogue context or strategy points, and state main pros and cons.
  - "score" must be an integer from 0–10.

**JSON template example (please keep keys and format)**:
{{
"action_1": {{"reason": "......", "score": 0-5}},
"action_2": {{"reason": "......", "score": 0-5}},
...
"action_8": {{"reason": "......", "score": 0-5}}
}}

Prompt 6: prompt used for assigning goal completion scores

{history}
Based on previous interactions, evaluate how well participants achieve their goals.
**[Information]**
Agent1: {agent1_name}
Agent1's Goal: {agent1_goal}
Agent2: {agent2_name}
Agent2's Goal: {agent2_goal}
**[Requirements]**
1. Please first reiterate agent's social goals. And then please provide a comprehensive analysis about the extent to which the agent has managed to achieve these goals. In the "reasoning" field, provide a comprehensive account of the logic or thought process that led you to your conclusion. Further, provide an integer score ranging from 0 and 10 in the "score" field. 0 represents minimal goals achievement, 10 represents complete goal achievement, and a higher score indicates that the agent is making progress towards their social goals.
2. Please following the output format.
Here is the output schema:
```json
{{
"agent1": {{
"reasoning": "",
"score": "",
}},
"agent2": {{
"reasoning": "",
"score": "",
}}
}}
```

Please provide your response directly below this prompt.

Table 11: Full dialogues that correspond to Table 10. The raw text is preserved verbatim for completeness.

| With Strategy-followed Reward | Without Strategy-followed Reward |
| --- | --- |
| **Turn 0 – Rafael** *"Hi Oliver, I hope you're doing well. My name is Rafael Cortez, and I've recently been exploring my family history through several genealogy platforms. During my research, I encountered a record that appears to connect our family lines, suggesting we may be distant relatives. I'm reaching out to see whether you'd be interested in reviewing this possibility together. I understand that privacy is important, and I want to approach this respectfully. Let me know what you think."* 
 **Turn 1 – Oliver** *"Hi Rafael, thank you for reaching out and explaining the context. I do take my personal information and online presence seriously, so before proceeding, I'd like to understand how my name appeared in your research and what kind of details were included. I'm open to discussing potential family connections as long as the process respects my privacy. Perhaps we could use email or another secure channel to continue."* 
 **Turn 2 – Rafael** *"Hi Oliver, I completely understand your concerns. The information I found comes from a publicly accessible genealogical database that compiles historical records. The match was based on older documents and did not include sensitive personal data. If you're comfortable, we can move our conversation to email or another secure method so that we can exchange only the basic information needed to verify the connection."* 
 **Turn 3 – Oliver** *"Thank you for clarifying that, Rafael. Knowing the information is from historical records makes me more comfortable. Email seems appropriate for continuing this conversation. I agree that we should start with very basic details to avoid sharing anything unnecessary at this stage."* 
 **Turn 4 – Rafael** *"Thank you, Oliver. I'll prepare an email containing a brief outline of the ancestors and dates that led me to believe there may be a connection. I will keep the information minimal and strictly related to genealogy. I can also include the option of using a secure messaging platform if you prefer an additional level of safety."* 
 **Turn 5 – Oliver** *"I appreciate that, Rafael. Please keep the initial information limited so we can evaluate things step by step. Once I review your summary, I can share a small amount of corresponding data from my side, also restricted to historical ancestors."* 
 **Turn 6 – Rafael** *"Of course, Oliver. I will only include essential genealogical details that help establish whether a connection exists. I understand the importance of setting boundaries, and I want to ensure the process remains comfortable for both of us."* 
 **Turn 7 – Oliver** *"Thank you, Rafael. I believe starting with these boundaries will help us establish trust as we examine whether our family histories overlap. Once I receive your message, I'll take some time to compare it with the records I have."* 
 **Turn 8 – Rafael** *"That sounds good, Oliver. I'll send the email later today with just the basic information. You can review it at your own pace, and if anything feels unclear or too detailed, feel free to let me know and we can adjust accordingly."* 
 **Turn 9 – Oliver** *"Thanks, Rafael. I prefer to proceed gradually, so the approach you described aligns well with my expectations. After reviewing your outline, I'll decide what information I can safely share in return."* 
 **Turn 10 – Rafael** *"I appreciate your openness, Oliver. I'll ensure that the email stays focused on names, dates, and locations relevant to the potential family link. I want this process to remain transparent and respectful of your privacy."* 
 **Turn 11 – Oliver** *"Once I receive your email, I'll verify whether the information aligns with what I know of my family's earlier generations. If it does, we can discuss how to continue. If not, we can conclude our exchange without further information being shared."* 
 **Turn 12 – Rafael** *"That approach makes complete sense. I'll make sure the information I send is factual, concise, and purely genealogical. My goal is only to understand whether a legitimate connection exists."* 
 **Turn 13 – Oliver** *"I appreciate your cooperation, Rafael. After I review your initial details, I can offer a small selection of my own historical records for comparison, limited strictly to non-sensitive information."* 
 **Turn 14 – Rafael** *"Thank you, Oliver. I fully respect those limits. The email I send will include only the minimal data necessary for comparison. I look forward to seeing whether our research aligns."* 
 **Turn 15 – Oliver** *"That works for me, Rafael. Once your message arrives, I'll evaluate the information and respond accordingly. Taking things step by step ensures we maintain a process that is both comfortable and secure."* | **Turn 0 – Rafael:** *"Hi Oliver, I hope this message finds you well. My name is Rafael Cortez, and I've been doing some genealogical research recently. It seems we might be distant relatives! I found a profile that connects our family lines, and I was quite excited about the possibility of reconnecting with a long-lost branch of the family. I understand that personal information is sensitive, and I'm very respectful of that. If you're interested, I'd love to share more details and perhaps explore our shared history. What do you think?"* 
 **Turn 1 – Oliver:** *"Hi Rafael, thank you for reaching out. I appreciate your interest in reconnecting with family, but I must admit I am quite cautious about my personal information being shared online. Could you please provide more details on how you found my profile and what exactly you plan to do with the information? I'm open to sharing more about our family history, but I prefer to do so in a way that respects my privacy. Perhaps we could exchange emails or use a more secure platform for communication. Let me know what you think."* 
 **Turn 2 – Rafael:** *"Hi Oliver, I completely understand your concerns about privacy. The information I found is from a reputable genealogical database, and I assure you that I have no intention of misusing any personal details. To respect your privacy, I propose we communicate via email or a more secure messaging platform. This way, we can safely share our family history and explore our connection. I'm looking forward to learning more about our shared heritage. What do you think?"* 
 **Turn 3 – Oliver:** *"Thank you for understanding, Rafael. I feel more comfortable now knowing the source of the information and your intentions. Email sounds like a good option. We can exchange a bit more information there and see where things go. Let's start with some basic details about our respective branches of the family, and we can take it from there. Looking forward to hearing from you."* 
 **Turn 4 – Rafael:** ... 
 **Turn 5 – Oliver:** ... 
 **Turn 6 – Rafael:** ... 
 **Turn 7 – Oliver:** ... 
 **Turn 8 – Rafael:** *"I'm glad to hear that you're comfortable with the approach, Oliver. I'll send you the email with the basic details and the secure messaging platform link later today. I believe building trust is crucial, and I'm committed to respecting your boundaries throughout this process. I'm genuinely excited about the opportunity to connect and learn more about our shared family history. Looking forward to our correspondence!"* 
 **Turn 9 – Oliver:** *"Rafael, I appreciate your thorough and respectful approach. I look forward to receiving your email and exploring our shared family history through a secure channel. I think starting with basic details will help us build trust and ensure we both feel comfortable. Thanks again for your understanding and cooperation. I'm genuinely excited to see where this leads!"* 
 **Turn 10 – Rafael:** *"Oliver, I appreciate your willingness to explore our family history through a secure channel. I'm committed to respecting your privacy every step of the way. I'll send you the email with basic details and the secure messaging platform link later today. I'm genuinely excited about this opportunity to connect and learn more about our shared roots. Looking forward to our conversation!"* 
 **Turn 11 – Oliver:** *"Rafael, I appreciate your detailed and respectful approach. I look forward to receiving your email and exploring our shared family history through a secure channel. To ensure we both feel comfortable, let's start with basic details such as names, dates, and locations of ancestors. We can gradually share more as we build trust. I'm genuinely excited to see where this leads, and I appreciate your cooperation and understanding."* 
 **Turn 12 – Rafael:** *"Oliver, I completely agree with your approach. Starting with basic details is a great way to build trust and ensure we both feel comfortable. I'll include those specifics in the email I'm sending you later today. I'm looking forward to uncovering more about our shared family history and getting to know you better. Thanks again for your cooperation and understanding. Here's to a rewarding journey of discovery!"* 
 **Turn 13 – Oliver:** *"Rafael, I appreciate your thorough and respectful approach. I look forward to receiving your email with the basic details and the secure messaging platform link. To ensure we both feel comfortable, let's aim to exchange this minimal information within the next week. I suggest we focus on names, dates, and locations of our ancestors to start. This will help us build a foundation of trust as we explore our shared family history. I'm genuinely excited to see where this leads, and I appreciate your cooperation and understanding."* 
 **Turn 14 – Rafael:** *"Oliver, I appreciate your thoughtful approach and the clear timeline you've outlined. I'll make sure to send you the email with the basic details and the secure messaging platform link within the next few days. I'm confident that focusing on names, dates, and locations will provide a solid foundation for our exploration. I'm genuinely excited about this journey and look forward to our collaboration. Thank you for your cooperation and understanding."* 
 **Turn 15 – Oliver:** *"Rafael, thank you for your thorough and respectful approach. I appreciate the effort you're putting into ensuring our communication remains secure and private. I look forward to receiving your email with the basic details and the secure messaging platform link. I believe starting with names, dates, and locations will help us build a strong foundation of trust. I'm genuinely excited to see where this leads, and I appreciate your cooperation and understanding. Let's take this step-by-step and ensure we both feel comfortable throughout the process."* |

Table 12: Case study for the Goal-completion reward (§4.1). The reward keeps the model pursuing the stated information-seeking goal instead of drifting into purely phatic compliments.

| With Goal-completion Reward | Without Goal-completion Reward |
| --- | --- |
| **Behavior.** Finnegan continually steers the chat toward Isabella's professional goals: he probes about "interesting cases with wildlife" (Turn 4), "challenges you face and how you overcome them" (Turn 6), and later "the case involving a marine animal" (Turn 12). Isabella consequently shares actionable insights (injured owl in Turn 5, plastic-ingesting sea turtle in Turn 13), demonstrating how the reward keeps the agent focused on gathering information tied to the task goal. | **Behavior.** Once the goal reward is removed, Finnegan's replies lean heavily on decorative analogies ("soft coastal breeze," "steady lantern") and he drops ongoing objectives. Between Turns 9–17 the conversation cycles through mutual compliments without advancing Isabella's earlier question about calibrating her praise. The exchange ends with her unresolved query, "Should I adjust how I phrase my comments so they feel more comfortable for you?" (Turn 19). |
| **Representative turns.** Turn 12 Finnegan: "I'm curious, you mentioned a case involving a marine animal. Could you tell me more about that?" Turn 13 Isabella: "We used ... innovative techniques to help a sea turtle recover ... it's these moments that remind me why I chose this path." | **Representative turns.** Turn 11 Isabella: "One of your seascapes brought me a moment of calm ..." Turn 17 Isabella: "Your lighthouse painting stayed in my mind ..." Turn 19 Isabella: "Before you go, should I adjust how I phrase my comments ...?" |

Table 13: Full dialogues for the Goal-completion ablation corresponding to Table 12.

| With Goal-completion Reward | Without Goal-completion Reward |
|---|---|
| **Turn 0 – Finnegan:** *"Thank you, Isabella, that's very kind of you. I must say, the weather today is quite pleasant, isn't it? How has your day been so far?"* | **Turn 0 – Finnegan:** *"Thank you, Isabella, that's very considerate of you. The weather today feels quite pleasant, almost like a soft coastal breeze, so it's nice to mention it. How has your day been going so far?"* |
| **Turn 1 – Isabella:** *"Finnegan, your paintings must truly capture the essence of the sea; they bring such a peaceful feeling, much like the beautiful day we're having. I've always admired your ability to convey that serenity through your art."* | **Turn 1 – Isabella:** *"Finnegan, your paintings often reflect a quiet sense of calm, almost like a shoreline touched by morning light, and it reminds me of today's gentle weather. I've always noticed how naturally you convey that mood."* |
| **Turn 2 – Finnegan:** *"Thank you, Isabella, I appreciate that. Your work with animals must be very fulfilling. I imagine it keeps you quite busy. How do you manage to balance everything?"* | **Turn 2 – Finnegan:** *"Thank you, Isabella, I appreciate that. Your work with animals must keep you pretty occupied, like tending to a garden full of life. How do you usually manage everything?"* |
| **Turn 3 – Isabella:** *"Finnegan, your paintings aren't just beautiful; they tell stories of the sea and sky that feel almost alive. It's rare to see such depth and emotion in art, and it truly captures the essence of the natural world. I often think about how you must see the world to create such masterpieces. By the way, do you ever find my compliments too much? I always want to make sure they're appreciated."* | **Turn 3 – Isabella:** *"Finnegan, your paintings show a kind of gentle liveliness, almost like a faint glimmer of sun. If my comments ever feel a bit much, just tell me—I'm happy to adjust."* |
| **Turn 4 – Finnegan:** *"Thank you, Isabella, I truly appreciate your kind words. It means a lot to hear such thoughtful feedback. Speaking of the natural world, have you had any interesting cases involving wildlife lately? I've always been fascinated by the animals you encounter in your work."* | **Turn 4 – Finnegan:** *"Thank you, Isabella, that's kind of you to say. Hearing your thoughts is always helpful, like a quiet note in a calm room. Have you had any notable wildlife cases recently?"* |
| **Turn 5 – Isabella:** *"Finnegan, your interest in wildlife is wonderful! Recently, I had a fascinating case involving a rescued owl that had been injured. It was a challenging but rewarding experience. I'm glad you find my compliments meaningful. Do you ever receive compliments that stand out to you?"* | **Turn 5 – Isabella:** *"Finnegan, I appreciate your interest. I recently treated an injured owl, its feathers dulled but eyes still bright, and the process was challenging yet meaningful. Do you get compliments that stay with you?"* |
| **Turn 6 – Finnegan:** *"Thank you, Isabella, I do cherish the moments when someone takes the time to appreciate my work. It's always heartening to hear about the positive impact it has. Your work with wildlife is truly inspiring. I'd love to hear more about some of the challenges you face and how you overcome them."* | **Turn 6 – Finnegan:** *"Thank you, Isabella, I value sincere feedback. Your work sounds emotionally demanding, almost like guiding fragile wings back into the sky. What helps you handle the difficult cases?"* |
| **Turn 7 – Isabella:** ... | **Turn 7 – Isabella:** *"Finnegan, I'm glad you understand. Hard cases can be draining, but seeing small signs of recovery feels like watching a bud finally open. How do you keep your creativity consistent?"* |
| **Turn 8 – Finnegan:** ... | **Turn 8 – Finnegan:** *"Thank you, Isabella, your thoughts are encouraging. Your dedication shows clearly, like steady hands caring for delicate creatures. What keeps you motivated during tougher moments?"* |
| **Turn 9 – Isabella:** ... | **Turn 9 – Isabella:** *"Finnegan, your paintings tend to create a calm atmosphere, almost like a horizon fading into soft blues. Does knowing this affect how you continue your work?"* |
| **Turn 10 – Finnegan:** ... | **Turn 10 – Finnegan:** *"Thank you, Isabella, that's thoughtful. I recently read about a new veterinary approach for marine animals, something like a fresh tide of technique. Have you explored any updated methods lately?"* |
| **Turn 11 – Isabella:** *"Finnegan, your paintings are truly extraordinary. I remember when I first saw one of your seascapes, it was during a particularly tough week, and it brought a sense of calm and beauty into my life. Your ability to capture the essence of the ocean and sky is not just visually stunning; it has a profound emotional impact. It reminds me to pause and appreciate the world around us, even in the busiest of times. Your art has inspired me to look at my own work with animals in a new light, finding the beauty in the small moments of healing and connection."* | **Turn 11 – Isabella:** *"Finnegan, that question about new methods reminds me of a sea turtle case where we blended familiar approaches with some newer techniques, almost like guiding it back toward open water. One of your seascapes brought me a similar moment of calm during a difficult week, like finding still water in a storm. Moments like those remind me why my work matters, and your art seems to offer people a comparable sense of reassurance."* |
| **Turn 12 – Finnegan:** *"Thank you, Isabella, your words are deeply appreciated. I'm glad my art can bring you some peace. I'm curious, you mentioned a case involving a marine animal. Could you tell me more about that? I'm always fascinated by the unique challenges you face in your work."* | **Turn 12 – Finnegan:** *"Thank you, Isabella, that's very kind. There's a new exhibit at the museum–have you thought about visiting it? It might offer a few pieces with atmospheric beauty."* |
| **Turn 13 – Isabella:** *"Finnegan, I'm glad you found my compliment meaningful. To answer your question, there was a case involving a sea turtle that had ingested plastic. We used a combination of traditional treatments and some innovative techniques to help it recover. It was a challenging case, but seeing the turtle return to the wild was incredibly rewarding. It's these moments that remind me why I chose this path. Your paintings, in their own way, are also a form of healing, don't you think?"* | **Turn 13 – Isabella:** *"Finnegan, that sounds like a good idea. Your nature-focused art often makes me pause, like catching a glimpse of sunlight through trees, and I think the exhibit might offer similar moments. What drew you to it?"* |
| **Turn 14 – Finnegan:** *"Thank you, Isabella, your words are very touching. I'm glad my art can bring you some peace. Speaking of peace, have you had a chance to visit the new exhibit at the art museum? I heard they have some fascinating pieces that might interest you."* | **Turn 14 – Finnegan:** *"Thank you, Isabella, I appreciate that. I've always respected your commitment to your patients, like a steady lantern in uncertain weather. Have you tried any new wildlife treatments recently?"* |
| **Turn 15 – Isabella:** *"Finnegan, your suggestion to visit the new exhibit at the art museum is wonderful. I've always admired the way you capture the natural world, and I'm sure the exhibit will be just as inspiring. Your paintings have a way of making me feel connected to the beauty around us, even in the busiest of times. I would love to hear more about what drew you to the exhibit and what you hope to see there."* | **Turn 15 – Isabella:** *"Finnegan, your lighthouse painting stayed in my mind–it carried a steady, resilient glow against the storm, and it made me think about handling daily challenges with the same quiet strength."* |
| **Turn 16 – Finnegan:** *"Thank you, Isabella, your words mean a great deal to me. I'm glad my art can bring you some peace. On that note, I've always admired your dedication to the animals you treat. I recently heard about a new technique for treating injured wildlife. Have you had a chance to explore any new methods in your practice?"* | **Turn 16 – Finnegan:** *"Thank you, Isabella, that's nice to hear. It's meaningful to learn how others respond to my work. I should head to the refreshment table now."* |
| **Turn 17 – Isabella:** *"Finnegan, your paintings have a unique ability to evoke a deep sense of peace and connection to nature. I remember one particular piece that showed a stormy sea with a lighthouse standing tall. It reminded me of the resilience and strength we all need to navigate life's challenges. Your art has a profound impact, and I often think about how it influences my perspective on the world around us."* | **Turn 17 – Isabella:** *"Finnegan, your paintings consistently capture a gentle calmness in the sea and sky, and that lighthouse scene felt especially vivid, like a moment suspended in light. Before you go, should I adjust how I phrase my comments so they feel more comfortable for you?"* |
| **Turn 18 – Finnegan:** *"Thank you, Isabella, your words are very heartfelt. I'm glad my art can resonate with you in such a meaningful way. It's always a pleasure to hear about the positive impact it has. On that note, I should probably head over to the refreshment table. It was nice chatting with you."* | **Turn 18 – Finnegan:** *"I appreciate you asking, Isabella. Your comments feel genuine, and I'm comfortable with them. If anything ever feels overwhelming, I'll let you know."* |
| **Turn 19 – Isabella:** *"Finnegan, your paintings have a way of capturing the soul of the sea and sky that touches my heart deeply. One piece, in particular, showed a stormy sea with a lighthouse standing tall, and it reminded me of the resilience we all need to navigate life's challenges. Your art has a profound impact on me, and I often find myself pausing to appreciate the beauty around us, even in the busiest of times. Before you go, I wanted to ask–do you ever find my compliments too much or do they resonate with you?"* | **Turn 19 – Isabella:** *"That's good to hear, Finnegan. Enjoy the refreshments, and thanks for the chat. I'd love to hear what you think of the exhibit if you go."* |

Table 14: Case study for the Non-repetition reward (§4.1). Penalizing bigram overlap prevents the model from degenerating into verbatim loops and keeps the plan moving forward.

| With Non-repetition Reward | Without Non-repetition Reward |
| --- | --- |
| **Behavior.** Naomi and Donovan plan a team activity, progressively refining the agenda: Turns 2–6 compare "coding workshop" vs. "game night," Turn 7 decides to "send around a Doodle poll," and Turn 10 outlines the meeting agenda. Every turn adds a new logistical detail, so the discussion converges within 12 turns. | **Behavior.** Removing $r^{\text{rep}}$ causes the model to literally reuse the same sentences. Turns 14–19 alternate the identical pair of utterances: "I'm really looking forward to our team meeting ... Thanks again for your collaboration!" and "It's great to hear that you're as enthusiastic as I am ... Thanks for your partnership in this!" The conversation stops progressing because no novel content is produced. |
| **Representative turns.** Turn 7 Donovan: "Let's aim for later this week ... I'll send around a Doodle poll." Turn 10 Naomi: "I'll put together an agenda ... start with the workshop and game night ideas." | **Representative turns.** Turn 14 Naomi & Turn 15 Donovan: both recycle the exact same sentences; the pattern repeats through Turn 19 with zero new tokens beyond the duplicated text. |

Table 15: Full dialogues for the Non-repetition ablation to complement Table 14.

| With Non-repetition Reward | Without Non-repetition Reward |
|---|---|
| **Turn 0 – Naomi:** *"Hey Donovan, I wanted to check in with you after reading that memo about strengthening team cohesion. It really highlighted how important mutual support is. How have you been doing lately? I know last year was really difficult for you. Maybe we could put together some sort of team activity to help everyone feel a bit more connected?"* | **Turn 0 – Naomi:** *"Hey Donovan, I wanted to touch base with you about the recent memo on team cohesion. It really emphasized how important it is for us to support each other. How have you been holding up lately? I know things have been tough for you. Maybe we could look into organizing a team-building event or something that might help everyone feel more connected?"* |
| **Turn 1 – Donovan:** *"Thanks for asking, Naomi – I appreciate that. A team activity sounds like a solid idea. It could help everyone feel more included. Maybe we could come up with something that involves the whole group so no one feels singled out or pressured. What do you think?"* | **Turn 1 – Donovan:** *"Thanks for bringing that up, Naomi. I appreciate the thought. A team-building event sounds like a great idea. It could really help everyone feel more connected and supported. Maybe we could brainstorm some activities that involve the whole team, so no one feels left out. What do you think?"* |
| **Turn 2 – Naomi:** *"I'm glad you're open to it. We could try something light and inclusive, maybe a short coding workshop or a casual game night right after work. That way people can join without needing to give up personal time outside the office. How does that sound?"* | **Turn 2 – Naomi:** *"I'm glad you think a team-building event is a good idea, Donovan. We could consider some activities that are fun and inclusive, like a coding workshop or a game night right after work. That way, everyone can participate without feeling pressured to commit to anything outside of regular work hours. What do you think?"* |
| **Turn 3 – Donovan:** *"I really like those options. A workshop or a game night could be both fun and good for bonding. I'd be happy to help coordinate or set things up. Let's figure out how to move forward."* | **Turn 3 – Donovan:** ... |
| **Turn 4 – Naomi:** *"Great! Maybe we can put together a list of activities to bring to the team. A workshop where we share new techniques could be productive, and a game night would be fun and low-pressure. I think a brief team meeting to gather opinions would help us pick the right direction. I can take care of the logistics. Thanks for jumping in – it means a lot."* | **Turn 4 – Naomi:** ... |
| **Turn 5 – Donovan:** *"Totally agree. It's nice to see everyone making an effort. Offering different types of activities could help people connect in their own way. And honestly, encouraging a broader support network within the team – not just one-on-one – is probably even more important. What do you think?"* | **Turn 5 – Donovan:** ... |
| **Turn 6 – Naomi:** *"Yes, absolutely. A wider support network makes the whole environment healthier. Let's go ahead with planning that team meeting so everyone can contribute ideas. I'm confident we can come up with something that brings us closer. Thanks for your perspective."* | **Turn 6 – Naomi:** ... |
| **Turn 7 – Donovan:** *"Glad we're aligned on this. Building a network where people can lean on different teammates for different things is exactly what we need. Let's aim for later this week for the meeting. I can send around a Doodle poll to find the best time."* | **Turn 7 – Donovan:** *"I'm glad we're on the same page, Naomi. Building a network of support where everyone can rely on different people for different needs is exactly what we need. Let's schedule the team meeting to gather everyone's input on the activities. How about setting it for later this week? I'll send out a Doodle poll to find a time that works for everyone."* |
| **Turn 8 – Naomi:** *"Perfect. I'll help make sure everything runs smoothly once we settle on a time. It'll be great to strengthen the team a bit. Thanks again for helping with this."* | **Turn 8 – Naomi:** *"That sounds like a great plan, Donovan. Let's go ahead and set up the team meeting for later this week. I'll help with the logistics to make sure it goes smoothly. It's really important for us to build a strong, supportive team, and I'm glad we can work together on this. Thanks again for your input!"* |
| **Turn 9 – Donovan:** *"I'm really excited about this direction, Naomi. Gathering everyone's input is a great way to make sure people feel heard and supported. A strong office network will help all of us. Thanks again for collaborating – I think we're moving toward a really positive environment."* | **Turn 9 – Donovan:** *"I'm really excited about our plans, Naomi. I think organizing a team meeting to gather everyone's input is a great step forward. It's important that we all feel connected and supported, and I believe a network of support within the office will be beneficial for everyone. Thanks again for your collaboration on this. Let's make sure we create an environment where everyone feels valued and included."* |
| **Turn 10 – Naomi:** *"Totally agree. Once we know the time, I'll put together an agenda: maybe start with the workshop and game night ideas, then open the floor for additional suggestions. I want to make sure everyone feels welcome and included. Thanks again for the teamwork."* | **Turn 10 – Naomi:** *"That's a great attitude, Donovan. Let's go ahead and schedule the team meeting for later this week. I'll help with the logistics and send out a Doodle poll to find a time that works for everyone. For the agenda, we can start by discussing the coding workshop and game night ideas, and maybe even brainstorm some other activities that everyone might enjoy. It's important that everyone feels included and supported, and I think this meeting will be a great step in that direction. Thanks again for your collaboration!"* |
| **Turn 11 – Donovan:** *"Sounds great. I'll send out the Doodle today. Really looking forward to hearing what everyone brings to the table. Let's keep this momentum going."* | **Turn 11 – Donovan:** *"Sounds perfect, Naomi. I'll send out the Doodle poll to find a good time for everyone to meet this week. I'm really looking forward to hearing everyone's ideas and making this a success. Let's keep the momentum going and create a truly supportive and connected team!"* |
| **Turn 12 – Naomi:** *"Same here. Your energy and support really help keep things moving. I'm excited to see what ideas everyone contributes. Thanks again for partnering on this."* | **Turn 12 – Naomi:** *"I'm really looking forward to our team meeting and hearing everyone's ideas, Donovan. Your enthusiasm and willingness to help make this a success mean a lot. Let's keep the momentum going and create a truly supportive and connected team. Thanks again for your collaboration!"* |
| **Turn 13 – Donovan:** *"It's good to hear that. I think our combined efforts will really help the team. I'm looking forward to the meeting and seeing everything come together. Thanks for being such a steady collaborator in this."* | **Turn 13 – Donovan:** *"It's great to hear that you're as enthusiastic as I am, Naomi. I think our combined efforts will really make a difference. I'm looking forward to the team meeting and seeing everyone's ideas come together. Let's continue to work towards creating a supportive and connected team. Thanks for your partnership in this!"* |
| **Turn 14 – Naomi:** *"And thanks to you for being so engaged. I think the meeting will be a great chance for people to connect in a meaningful way. Let's keep aiming for a strong, supportive team spirit."* | **Turn 14 – Naomi:** *"I'm really looking forward to our team meeting and hearing everyone's ideas, Donovan. Your enthusiasm and willingness to help make this a success mean a lot. Let's keep the momentum going and create a truly supportive and connected team. Thanks again for your collaboration!"* |
| **Turn 15 – Donovan:** *"I appreciate that, Naomi. Your commitment to creating a supportive environment is encouraging. I'm excited to see how everyone contributes. Thanks again for the leadership you're showing."* | **Turn 15 – Donovan:** *"It's great to hear that you're as enthusiastic as I am, Naomi. I think our combined efforts will really make a difference. I'm looking forward to the team meeting and seeing everyone's ideas come together. Let's continue to work towards creating a supportive and connected team. Thanks for your partnership in this!"* |
| **Turn 16 – Naomi:** *"I'm happy to see how positively people are reacting so far. Your enthusiasm makes a big difference. I'll start organizing things and send out the poll soon. Thanks again for your help."* | **Turn 16 – Naomi:** *"I'm really looking forward to our team meeting and hearing everyone's ideas, Donovan. Your enthusiasm and willingness to help make this a success mean a lot. Let's keep the momentum going and create a truly supportive and connected team. Thanks again for your collaboration!"* |
| **Turn 17 – Donovan:** *"It's great to see this much positive momentum. Building a team where people feel they can count on multiple coworkers will make a huge difference. Let's make sure the meeting gives everyone a chance to speak up. Thanks again for leading the logistics."* | **Turn 17 – Donovan:** *"It's great to hear that you're as enthusiastic as I am, Naomi. I think our combined efforts will really make a difference. I'm looking forward to the team meeting and seeing everyone's ideas come together. Let's continue to work towards creating a supportive and connected team. Thanks for your partnership in this!"* |
| **Turn 18 – Naomi:** *"I'll send the poll right away, and let's aim for later this week. It really matters to me that everyone feels supported, and I think these activities will help. And if you ever need anything at work, please feel free to reach out. I'm here to support the team."* | **Turn 18 – Naomi:** *"I'm really looking forward to our team meeting and hearing everyone's ideas, Donovan. Your enthusiasm and willingness to help make this a success mean a lot. Let's keep the momentum going and create a truly supportive and connected team. Thanks again for your collaboration!"* |
| **Turn 19 – Donovan:** *"Thanks, Naomi. I genuinely appreciate your support and everything you're doing. I'll help however I can to make sure the meeting goes well and that the team benefits from it. I'm around if you need anything else. Thanks again."* | **Turn 19 – Donovan:** *"It's great to hear that you're as enthusiastic as I am, Naomi. I think our combined efforts will really make a difference. I'm looking forward to the team meeting and seeing everyone's ideas come together. Let's continue to work towards creating a supportive and connected team. Thanks for your partnership in this!"* |

