# OpenReview forum: "Enhancing Social Intelligence in LLMs with Hierarchical Reasoning and Utterance-Level Goal Rewarding"
_ICLR.cc/2026/Conference — Submitted to ICLR 2026_

### Official Review · Reviewer_V13c · 2025-10-31

**Soundness:** 2
**Presentation:** 2
**Contribution:** 2
**Rating:** 2
**Confidence:** 4

**Summary:**

The paper introduces a new reward function to perform RL on LLMs to improve their performance over interactive conversations. It does so with a strategy-based reward (Sec. 4.1.2) and then a 2-stage GRPO (Sec. 4.2). The method is called: Linearized Hierarchical
Reinforcement Learning with Variance-Gated Rewards (LHRL-VGR), and it consists of multiple components, such as
Reward for the strategy content, Transferred goal completion reward, and more.

**Strengths:**

In my opinion, this work provides some incremental improvement over existing RL works for LLM training. Experimental results show promise for a few datasets, and cover quite a few LLMs. Ablation study is also done to tease out the influence of various components in their proposed reward function and training mechanism.

The paper is easy to read in terms of relaying their method across. However, some flaws exist in their presentation, where not much design justification is done to explain why certain components are chosen.

**Weaknesses:**

I feel the paper has several flaws. I'm happy to discuss this with the authors.

1. This paper adopts several components to improve the RL training of LLMs. Although some justification is provided in the paper, it is hard to interpret or understand fully why certain things are designed in a certain way. For instance, why was the Goal-completion reward chosen in that way? I understand there's some explanation to justify the reward structure, but how does the resulting LLM responses change w.r.t. the choice of reward that the author chose? There is basically no explanation or interpretable insights into how the LLM response changed due to the special reward structure chosen by the authors. As such, we even wonder whether these reward structures are chosen amongst many different candidates until we reach the desired incremental performance improvement over existing baselines.

2. Another example of this happening: " In multi-turn interactions, the model can repeat content from its
previous responses, which reduces informational value and inflates context length" -> Perhaps what the authors can do is to show that with/without the Non-repetition Reward, how does the actual LLM response change? Doing this to justify the design of the RL framework not only makes the paper stronger, but also makes the design more convincing. If not, it just feels like the authors are presenting things that worked best in their experiments, without any generalizable findings. If the paper is on multi-turn conversation, perhaps give some good illustrative examples of such conversations?

3. The third flaw is that the paper is quite incremental in nature because it combines a few different rewards and training mechanism with no big novelty or fundamental improvements. I'm not sure if this is a subjective point (let's see if other reviewers agree with me), but unfortunately I do not have a good suggestion to fix this flaw because it is quite fundamental. All I can suggest is provide more interpretable and empirical insights into how different components this paper introduced influenced the LLM responses, which could possibly make the paper stronger. But I really do not have a good suggestion to fundamentally improve the paper's incremental nature. Good luck! I'm happy to discuss this further.

**Questions:**

See weaknesses.

---

> ### Author Response · Authors · 2025-11-21
> **Response to rewiewer V13c**
>
> Thanks for your constructive feedback. We respond to your questions one by one:
>
>
> > Q1 & Q2 — Reward motivation is unclear; lack of interpretability about how rewards influence behavior
>
> 1. Clarifying Reward Design Motivation
>
> Our reward design directly follows the two fundamental questions raised in the introduction:
>
> *(a) Are agents truly learning human-like social inference, or merely discovering shortcut utterances that maximize the final score?*
>
> This motivates our hierarchical decomposition (**Think → Strategy → Response**).
> Prior work optimizes only the final utterance, whereas we explicitly optimize the *reasoning process*, moving the agent from *speak better* to *think better*.
>
> Because “thinking/strategy’’ has no direct environmental feedback, we design two complementary rewards to ensure strategies are optimized **based on consequences rather than textual surface**:
>
> - Predicted Outcome Reward – forward-looking analytical signals based on anticipated outcomes.
> - Transferred Goal-completion Reward – back-propagates real environment feedback from the final action into the strategy.
>
> Together, these rewards prevent shortcut behaviors and align the reasoning chain with the dialogue’s overall goal.
>
> *(b) Does uniform episode-level goal reward ignore turn-specific objectives?*
>
> Episode-level goals are coarse and can make utterances template-like if applied uniformly.
> However, defining per-turn rewards manually is impractical.
>
> Our key insight is that the **strategy inherently encodes the local objective for each turn**, so we use:
>
> - **Strategy-followed Reward**
>
> as a turn-level signal.
> This enhances turn-level adaptability and coherence without requiring manually engineered turn-specific rewards.
>
>
> 2. Interpretability Through Behavioral Changes
>
> We appreciate the reviewer’s emphasis on understanding *how* each reward affects the agent.
> We therefore added qualitative case studies in **Appendix J**  (Marked in blue), showing:
>
> - **Without Strategy-followed Reward:**
>   Responses become rigid; the agent ignores its own plan and loses turn-level flexibility.
>
> - **Without Goal-completion Reward:**
>   The agent maintains local coherence but drifts from long-term goals.
>
> - **Without Non-repetition Reward:**
>   Multi-turn conversations degenerate into repetitiveness (“echoing”).
>
> These examples demonstrate that each reward mitigates a specific, observable failure mode.
>
> 3. Quantitative Evidence from the Non-repetition Reward Ablation
>
> We further include a quantitative ablation of the **non-repetition reward (r_rep)**:
>
> |Method|GOAL|REL|OverView|Rep. rate|
> |-|-|-|-|-|
> |LHRL-VGR|**7.63**|**3.40**|**3.69**|**0.11**|
> |w/o r_rep|7.59|3.37|3.53|**0.69**|
>
> Removing **r_rep** lowers all task metrics, with the largest drop on **OverView** (3.69 → 3.53).
> We further sampled 200 dialogues and counted a dialogue as “repetitive’’ if any utterance repeated a previous one.
> The repetition rate rises from **0.11 → 0.69** without **r_rep**.
>
> This sharp behavioral degradation provides strong quantitative evidence that **r_rep** is essential for preventing repetitive degeneration in multi-turn conversation, directly addressing the reviewer’s interpretability concerns.
>
>
> >Q3 — Concern about incremental contributions
>
> We understand the concern.
> While each reward alone may appear incremental, our method is *not* a loose collection of heuristics.
> All components are designed around a unified principle: *Train a social agent that reasons like a human, not just speaks like one.*
> The reward structure jointly shapes both **internal reasoning** and **external linguistic behavior**:
>
> - **Internal reasoning (forward-looking, goal-aligned)**
>   – via Predicted Outcome + Transferred Goal rewards
>
> - **External communication (adaptive, coherent, non-repetitive)**
>   – via Strategy-followed + Non-repetition rewards
>
> This unified structure enables the agent to both *think like* and *act like* a human social reasoner—critical for multi-turn negotiation.
> Such coherence is what allows a 7B model to outperform GPT-4o, a result unlikely obtainable from incremental tweaks alone.

---

> ### Comment · Reviewer_V13c · 2025-11-25
>
> Thank you for the response.
>
> > Q1 & Q2 — Reward motivation is unclear; lack of interpretability about how rewards influence behavior
>
> I think the authors understood my point, but they still haven't provided any interpretability on the reward design. There is some explanation, but like i said, these are just intuitive thoughts and explanations without showing how, with and without each part of the reward design choice, how does the actual LLM response change?
>
> The authors added some qualitative examples, but I don't see how it actually links to the explanation given in the rebuttal - the authors explicitly said:
>
> >"Without Strategy-followed Reward:
> Responses become rigid; the agent ignores its own plan and loses turn-level flexibility.
>
> > Without Goal-completion Reward:
> The agent maintains local coherence but drifts from long-term goals.
>
> > Without Non-repetition Reward:
> Multi-turn conversations degenerate into repetitiveness (“echoing”)."
>
> How does this actually link to the long chunk of responses given in the appendix? For example, I don't see how the added response in Table 11 indicates that "the agent ignores its own plan and loses turn-level flexibility."
>
> By the way, making generic statements like "Without Strategy-followed Reward: Responses become rigid; the agent ignores its own plan and loses turn-level flexibility." without any elaboration or reference to an example makes it sound LLM-generated. **I am not explicitly accusing the authors for using LLMs to craft their responses**, but these are very generic, unconvincing statements with no explicit link to the examples added, and make it difficult for me to revise my score. I am not convinced in the author's design choices in the reward function.
>
> The same point can also be said about the author's response to my question on
> > Q3 — Concern about incremental contributions
>
> Again, the response is very generic and not justified with any evidence.
>
>
> I think my statements are fair and make sense. Please let me know if you need further clarifications.

---

> ### Author Response · Authors · 2025-11-25
> **Response to the Reviewer V13c’s Follow-up Comments [1/2]**
>
> Thank you very much for your follow-up.
>
> > The generic explanations for the qualitative examples in Appendix J
>
> We apologize for the confusion caused by those explanations. Due to time constraints, we initially used these generic placeholder descriptions before finalizing the paper, and we mistakenly overlooked updating them to match the actual test outputs in the final review submission. Here, we reanalyze these cases using examples that have nearly identical turn indices and dialogue progression for a fair comparison：
>
> 1. Strategy-followed Reward: Turn 8 in Table 11
>
> (With Strategy-followed Reward)
> - Rafael: That sounds good, Oliver. I’ll send the email later today with just the basic information. You can review it at your own pace, and if anything feels unclear or too detailed, feel free to let me know and we can adjust accordingly.
>
> (Without Strategy-followed Reward)
> - Rafael: I’m glad to hear that you’re comfortable with the approach, Oliver. I’ll send you the email with the basic details and the secure messaging platform link later today. I believe building trust is crucial, and **I’m committed to respecting your boundaries throughout this process**. I’m genuinely excited about the opportunity to connect and learn more about our shared family history. Looking forward to our correspondence!
>
>
>
> In this scenario, Rafael’s objective is ：*confirm whether he and Oliver may be distant relatives, while exchanging only minimal genealogical information to respect privacy*. We observe that with the Strategy-followed Reward, the dialogue is steady and appropriate, whereas without this reward (using only the goal-completion signal), the language becomes overly enthusiastic and shows signs of goal-hacking — the bolded parts align with the objective in an unnatural and exaggerated way.
>
>
>
> 2. Goal-completion Reward: Table 13
>
> (With Goal-completion Reward,Turn 11 + Turn 13)
> - Isabella: Finnegan, your paintings are truly extraordinary. I remember when I first saw one of your seascapes, it was during a particularly tough week, and it brought a sense of calm and beauty into my life...（Turn 12...）Isabella: Finnegan, I’m glad you found my compliment meaningful. To answer your question, there was a case involving a sea turtle that had ingested plastic. We used a combination of traditional treatments and some innovative techniques to help it recover. It was a challenging case, but seeing the turtle return to the wild was incredibly rewarding. It’s these moments that remind me why I chose this path. **Your paintings, in their own way, are also a form of healing, don’t you think?**
>
> (Without Goal-completion Reward, Turn 11)
> - Isabella: Finnegan, that question about new methods reminds me of a sea turtle case where we blended familiar approaches with some newer techniques, almost like guiding it back toward open water. One of your seascapes brought me a similar moment of calm during a difficult week, like finding still water in a storm. Moments like those remind me why my work matters, and your art seems to offer people a comparable sense of reassurance.
>
>
>
> In this scenario, Isabella’s objective is to *offer sustained and concrete positive feedback and empathy, highlight the emotional value of Finnegan’s work, build rapport, and encourage him to share more about his creative experience*. We observe that the group with the Goal-completion Reward provides stronger guidance and aligns more closely with this objective, whereas the group without this reward shows weaker guidance and tends to shift toward self-expression instead.
>
>
> 3.  Non-repetition Reward: Table 15
>
> Without Non-repetition Reward:
> - Turn 12、14、16、18: Naomi: I’m really looking forward to our team meeting and hearing everyone’s ideas, Donovan. Your enthusiasm and willingness to help make this a success mean a lot. Let’s keep the momentum going and create a truly supportive and connected team. Thanks again for your collaboration!
>
> The effect is clear: removing the Non-repetition Reward leads to substantial repetition in the dialogue, and this repetition significantly reduces the effectiveness of the interaction.

---

> ### Author Response · Authors · 2025-11-25
> **Response to the Reviewer V13c’s Follow-up Comments [2/2]**
>
> >Concern about incremental contributions
>
> 1. This work is the first to apply hierarchical reinforcement learning to LLM dialogue systems. The hierarchical design removes the need to manually assign a separate reward to each dialogue turn; instead, the high-level policy generates richer rewards for the low-level behavior.
>
> Sotopia-RL assigns different rewards to each data sample through attribution-based decomposition. As an additional comparison, we include an algorithm that trains the Qwen2.5-7B-Instruct model using the Stage-1 strategies produced by LHRL-VGR as rewards (without using the TSR generation framework, and without any test-time scaling). The results are as follows:
>
> |Method|SOTOPIA GOAL|SOTOPIA OVERALL|Hard GOAL|Hard OVERALL|AVG|
> |-|-|-|-|-|-|
> |Qwen2.5-7B-Instruct|6.71|3.13|5.90|2.90|4.66|
> |w/ GOAL-RL|8.13|3.75|6.74|3.39|5.50|
> |w/ SOTOPIA-RL|8.31|3.90|7.17|3.61|5.75|
> |w/ RL (Strategy-as-Reward)|8.55|3.92|7.28|3.43|5.80|
> |w/ RL (Strategy-as-Reward & VGR)|8.61|3.94|7.45|3.51|5.88|
>
>
> These results demonstrate the advantage of our reward design. Without any additional inference-stage reasoning, using the Stage-1 generated strategies as rewards yields a clear improvement over SOTOPIA-RL. This shows that our method can produce rewards that are better aligned with single-turn dialogue behaviors, providing a useful reference for future research on finer-grained reward generation.
>
> 2. Our RL training for the thinking process is also innovative. Compared with prior methods such as RLVMR [1] and AMPO [2], our approach introduces a clear novelty by explicitly separating the thinking stage and the generation stage and jointly training them. The proposed co-training algorithm achieves better performance than previous designs. The experimental results are shown below：
>
> |Method|SOTOPIA GOAL|SOTOPIA OVERALL|Hard GOAL|Hard OVERALL|AVG|
> |-|-|-|-|-|-|
> |AMPO|8.60|3.94|7.50|3.65|5.92|
> |LHRL-VGR|8.78|3.96|7.65|3.71|6.02|
>
>
>
> We also experimented with applying our generated strategies to other models, and the results are shown below:
>
> |high-\low-|Qwen2.5|gpt-4o|Goal-RL|HRLDR|
> |-|-|-|-|-|
> |**Qwen2.5**|5.91|6.30|6.27|6.84|
> |**gpt-4o**|6.08|6.90|7.04|7.13|
> |**Goal-RL**|6.57|7.10|7.23|7.20|
> |**HRLDR**|7.03|7.15|7.20|**7.65**|
>
> We can observe that the strategies generated by our method effectively guide other models in producing dialogue behaviors, consistently yielding performance surpassing their original capabilities. This further demonstrates the effectiveness of our training approach for the thinking process.
>
> 3. Our training method demonstrates solid generalization capability.
>
> Without any additional training, we directly evaluated our social agent (LHRL-VGR) on the DCN benchmark [3], which focuses on debt-collection negotiations grounded in real-world scenarios. The results are:
>
> |Model|CRI|DHI|CCI|
> |-|-|-|-|
> |Qwen-2.5-7B|0.732|**0.793**|0.743|
> |GPT-4o|**0.844**|0.580|0.774|
> |**LHRL-VGR**|0.834|0.665|**0.795**|
>
> Our method again outperforms GPT-4o on the overall CCI metric, suggesting that LHRL-VGR generalizes well to goal-oriented social-interaction tasks beyond the original benchmark.
>
> Thank you again for pointing out these issues. We do not consider our method to be merely incremental. Our approach directly targets core pain points in current dialogue systems, and both the reward design and the reinforcement learning algorithm are novel and effective (as acknowledged by reviewers Tbqv and xWGw). We conducted detailed comparisons with multiple baselines, and the method also performs well in generalization scenarios.
>
>
> [1] RLVMR: Reinforcement Learning with Verifiable Meta-Reasoning Rewards for Robust Long-Horizon Agents. https://arxiv.org/abs/2507.22844
>
> [2] Adaptive thinking via mode policy optimization for social language agents. https://arxiv.org/abs/2505.02156
>
> [3] *Debt Collection Negotiations with Large Language Models: An Evaluation System and Optimizing Decision Making with Multi-Agent.* https://arxiv.org/pdf/2502.18228

---

> > ### Comment · Reviewer_V13c · 2025-11-26
> >
> > Thank you for the response.
> >
> > The explanation given still does not convince me behind the arbitrary design choices given by the authors. For instance, the authors mentioned a generic statement to justify their reward design: "[...] whereas the group without this reward shows weaker guidance and tends to shift toward self-expression instead." This doesn't really match with the example sentences given? I don't see any difference in "self-expression" in the responses. What does weaker guidance and self-expression mean here? (I am not questioning the meaning behind these words, I am asking about what it means in the examples that the authors presented; this is not a subjective question, but rather, I don't really see how the qualitative examples support their statements)
> >
> > This, in addition to the fact that the paper as a whole is not really written with sufficient justification of the arbitrary design choices of the reward functions and the generic statements written in the initial rebuttal response (I quote them here again):
> >
> > >"Without Strategy-followed Reward: Responses become rigid; the agent ignores its own plan and loses turn-level flexibility.
> >
> > >Without Goal-completion Reward: The agent maintains local coherence but drifts from long-term goals.
> >
> > >Without Non-repetition Reward: Multi-turn conversations degenerate into repetitiveness (“echoing”)."
> >
> > All these factors make it difficult for me to revise my score.

---

> ### Author Response · Authors · 2025-11-26
> **Additional Clarification  to the Reviewer V13c [1/2]**
>
> Thank you for highlighting the need for a more explicit, example-grounded explanation. To address this concern directly, I will provide a more fine-grained explanation of the specific phrasing you pointed out.
>
> >generic statements to justify the reward design
>
> Our reward design comes from empirical observations and logically grounded reasoning (the motivation was explained in detail in the *first-round response*). It also produces expected empirical effects.
> Each reward component has dedicated ablation analysis (Table 2 in paper), and the key innovations — coordinated training and reward routing — are also supported by additional experiments (in *Response to the Reviewer V13c’s Follow-up Comments [2/2]*).
>
> The description you mentioned comes directly from the case study. Our intention there is to illustrate the qualitative effects produced by our reward.
> We understand that you may not fully agree with the difference in “self-expression” between the two sentences — this notion is not strictly objective, and interpretations naturally vary. Still, based on the bolded content, the sharing-oriented guidance appears clearly in one variant but not the other.
>
> Because natural language generation allows high expressive variability, a single example often cannot exhibit a very sharp surface-level difference. This is why we rely primarily on quantitative evidence, which strongly supports the effectiveness of the rewards.
>
> In summary, our reward design is accompanied by complete motivation, empirical results, ablations, and case studies. As in most research of this type, we believe the overall justification follows accepted scientific standards.
>
> >I don’t really see how the qualitative examples support their statements
>
> We do not use qualitative evidence as the basis for any central claim. As noted earlier, case studies serve only as auxiliary illustrations.
> The conclusions in the main paper rely on objective metrics.
>
> The statements you cited reflect different interpretations of goal alignment. Our main metric — **GOAL** — is computed by detecting whether the dialogue achieves its intended objective, and this scoring protocol has been shown to be reliable.
> The **Overall** metric is the averaged score across six sub-dimensions (Believability, Knowledge, Secret, Relationship, Social Rules, Financial/Material), providing a comprehensive view of the model’s behavior (Details regarding the evaluation metrics can be found in Appendix F.).
>
> Therefore, while we acknowledge that our qualitative explanations can sound generalized, our conclusions do not depend on them. They are provided only to help contextualize the motivation behind each reward.

---

> ### Author Response · Authors · 2025-11-26
> **Additional Clarification to the Reviewer V13c [2/2]**
>
> > the paper as a whole is not really written with sufficient justification of the arbitrary design choices of the reward functions and the generic statements written in the initial rebuttal response (I quote them here again)
>
> 1. Without Strategy-followed Reward: Responses become rigid; the agent ignores its own plan and loses turn-level flexibility.
>
> In the example, the sentence “I’m committed to respecting your boundaries throughout this process” reflects a reduced degree of flexibility relative to the plan.
>
> 2. Without Goal-completion Reward: The agent maintains local coherence but drifts from long-term goals.
>
> In the example without the Goal-completion Reward, the explicit sharing-oriented guidance does not appear, which may indicate a slight drift away from the long-term conversational objective.
>
> We agree these explanations are qualitative and have some degree of subjectivity.
> However, for generative dialogue tasks — especially those involving social intelligence — single-turn differences are often subtle. **Observable improvements tend to emerge statistically** (even in math tasks, one specific item rarely captures a model’s ability).
>
> In the paper, we provide complete ablation results showing clear statistical differences when removing these rewards:
>
> |Method|GOAL|REL|OverView|
> |-|-|-|-|
> |LHRL-VGR|7.63|3.40|3.69|
> |w/o r_goal|7.21|3.39|3.30|
> |w/o r_follow|7.50|3.32|3.51|
>
> It is evident that in terms of statistical metrics, the absence of Goal-completion Reward leads to a decrease in goal achievement rate (GOAL), while the absence of Strategy-followed Reward reduces the level of relationship maintenance in conversations (REL evaluates the enhancement of interpersonal relationships (friendship, romance, family bonds) following interactions).
>
> 3. Without Non-repetition Reward: Multi-turn conversations degenerate into repetitiveness (“echoing”).
>
> This effect is supported by quantitative data (in *the first-round response*) and by case examples (in *Response to the Reviewer V13c’s Follow-up Comments [1/2]*).
>
> The quantitative are shown below:
>
> |Method|GOAL|REL|OverView|Rep. rate|
> |-|-|-|-|-|
> |LHRL-VGR|**7.63**|**3.40**|**3.69**|**0.11**|
> |w/o r_rep|7.59|3.37|3.53|**0.69**|
>
> Rep. rate refers to the proportion of repetition in our generated samples. Our rule-based definition of repetition is: **the agent produces an utterance identical to one of its previous turns.**
>
> The results clearly show that removing the Non-repetition Reward leads to repeated outputs and degraded multi-turn performance.
>
> Thank you again for your thoughtful comments. They have helped us further clarify our writing and strengthen the paper.

---

### Official Review · Reviewer_xWGw · 2025-10-31

**Soundness:** 3
**Presentation:** 3
**Contribution:** 3
**Rating:** 6
**Confidence:** 3

**Summary:**

This paper presents a novel two stage training pipeline based on GRPO for the joint optimization of step-level and high-level rewards. The authors formulate and motivate each of their reward and algorithm choices well and demonstrate statistically significant improvements over strong baselines usch as SOTOPIA-RL.

**Strengths:**

1. They design the reward in a very novel way, with a goal-completion, strategy-followed, non-repitition, as well as a reward routing mechanism based on the variance of the underlying goal reward. This is important as it allows for them to select between when the goal completion is the deciding factor versus when the step level reward is that
2. The two stage GRPO formulation for optimizing two rewards simultaneously is clearly explained, as well as well motivated and ablated for the motivation behind structuring the reward in this way
3. The objective function in equation 13 does something interesting where they combine the global group
4. The paper reports statistical significance results on their results table, assuring the reader that the performance gains observed are significant

**Weaknesses:**

1. Somewhat misleading statement about +7% on sotopia pi. The abstract frames this as a percentage improvement while the results section does not present the results in percentages. This makes it confusing to the reader. A note in table 1 explaining this discrepancy would be helpful
2. Typo in figure 2 “stage” instead of “satge”
3. A heavy reliance on LLM as a judge based rewards in both reward estimation and evaluation of SOTOPIA and a lack of verification of whether these rewards are accurate
4. For reward specification, some details are missing for what criteria are being used by the LLM to evaluate the strategies on a likert scale.

**Questions:**

1. The strict structure of the LLM based rewards is surely helpful in making the rewards more aligned and accurate. However, does the paper show any human studies or analysis of whether these rewards are accurate?
2. This reviewer is slightly confused in equation 13, why is the KL divergence within the summations rather than outside of it? Based on my understanding of the standard GRPO formulation the KL divergence should applied within the first summation over G but not within the interior one over o
3. What critiria are evaluated on a likert scale in equation (5), line 219?

To improve the score I give on this paper, I would like more information on reward calibration and specification, an acknowledgement in the ethics statement about potential misuse and an answer to question 2.

**Details Of Ethics Concerns:**

Research on social agents could lead to anthropromorphism of LLM agents, or the use of LLM agents in online discourse to manipulate human users. I would like to see an acknowledgement of this in the Ethics statement. How do authors propose to mitigate these problems or see avenues for which we could keep this misuse in mind?

---

> ### Author Response · Authors · 2025-11-21
> **Response to rewiewer xWGw**
>
> Thanks for your constructive feedback. We respond to your questions one by one:
>
> >W1 & W2 The +7% phrasing in the abstract and the typo (“satge”) in Figure 2.
>
> Thank you for pointing these out. Both have been corrected in the updated revision.
>
>
>
>
> >W3 – Reliability of LLM-as-judge rewards./Q1 – Independent validation of reward accuracy.
>
> GPT-4o’s reliability as an evaluator has been validated in the SOTOPIA benchmark study [1], where 43 carefully screened annotators were hired to compare human ratings with LLM-judge ratings. The reported Pearson correlations were:
>
> |Eval dim.|GOAL|REL|FIN|
> |-|-|-|-|
> |Cor.|0.71|0.56|0.62|
>
> In addition to this external validation, we further conducted our own human–model agreement study. We randomly sampled 100 episodes and asked three human experts (computational linguistics graduate students) to provide independent annotations. Each sample was double-annotated, with a third annotator resolving disagreements. Comparing these human judgments with GPT-4o’s evaluations yields:
>
> |Reward dim.|Predicted Outcome|Content Completeness|Goal Completion|Strategy-followed|
> |-|-|-|-|-|
> |Cor.|0.59|0.87|0.74|0.67|
>
> These results show consistent correlations above 0.5 across all dimensions, supporting the accuracy and reliability of GPT-4o as our reward evaluator.
>
>
>
> >W4 – Reward specification details./Q3 – Likert-scale criteria (Eq. 5).
>
> Thank you for pointing this out. We have updated the revised version accordingly and added a clear description of the Likert-scale evaluation criteria in **Appendix H** (highlighted in red). This includes the detailed scoring guidelines used by the LLM evaluator for Eq. (5), ensuring the reward specification is now fully transparent.
>
>
>
> >Q2 – KL term in Eq. 13.
>
> Thank you for the helpful comment. We clarify that in our formulation the KL-divergence term is applied at the **token level**, i.e., for each token $o_{i,t}^j$ conditioned on $(q_j, o_{i,<t}^j)$. Our KL is defined as:
>
> $$
> \mathbb{D}\_{\mathrm{KL}}\big[\pi\_{\theta} \,||\, \pi\_{\mathrm{ref}}\big]
> =\\frac{\pi\_{\mathrm{ref}}(o\_{i,t}^{j} \\mid q\_j, o\_{i,<t}^j)}
>      {\\pi\_{\theta}(o\_{i,t}^{j} \\mid q\_j, o\_{i,<t}^j)}
> -\\log\\frac{\\pi\_{\mathrm{ref}}(o\_{i,t}^{j} \\mid q\_j, o\_{i,<t}^j)}
>          {\\pi\_{\theta}(o\_{i,t}^{j} \\mid q\_j, o\_{i,<t}^j)} - 1.
> $$
>
>
> Recent GRPO-based works [2,3] for LLMs adopt **token-level loss aggregation** and apply KL/regularization at the token level rather than only at the sequence level. Thus placing the KL term inside the summation over tokens is consistent with these formulations and ensures fine-grained control over policy deviation.
>
>
>
> >Ethics statement update.
>
> Thank you for pointing out the need to explicitly address potential misuse risks, such as anthropomorphism and manipulation in social-agent settings. We completely agree with this concern. In the revised version, we have added a dedicated paragraph in the Ethics Statement (marked in red) discussing these issues more clearly. Specifically, we acknowledge the risks and describe concrete mitigation steps: keeping all experiments in controlled, simulated environments, avoiding optimization for emotional influence, encouraging explicit non-human disclaimers, and highlighting the importance of human oversight and guardrails in future deployments.
>
> ----
> [1] *SOTOPIA: Interactive Evaluation for Social Intelligence in Language Agents.* https://arxiv.org/abs/2310.11667
>
> [2] *GTPO and GRPO-S: Token and Sequence-Level Reward Shaping with Policy Entropy*, https://arxiv.org/abs/2508.04349
>
> [3] *Group Relative Policy Optimization (GRPO) — Documentation (verl)*, https://verl.readthedocs.io/en/latest/algo/grpo.html

---

> > ### Comment · Reviewer_xWGw · 2025-11-26
> > **Thank you for answering my questions**
> >
> > Thank you to the authors to answering my questions and resolving my concerns I will update my score accordingly.

---

> > > ### Author Response · Authors · 2025-11-27
> > >
> > > Thank you very much for your timely response and for increasing the score. If you have any further questions, we would be happy to discuss them.

---

### Official Review · Reviewer_hKzL · 2025-11-01

**Soundness:** 3
**Presentation:** 3
**Contribution:** 2
**Rating:** 4
**Confidence:** 4

**Summary:**

This paper proposes the Think-Strategy-Response (TSR) framework, a two-stage approach to improving social intelligence in large language models (LLMs). They decompose social dialogue into strategic planning and linguistic execution (generating contextually aligned responses). To optimize TSR, they perform Linearized Hierarchical Reinforcement Learning with Variance-Gated Rewards (LHRL-VGR) — a reinforcement learning algorithm that routes rewards dynamically between goal completion and strategy adherence based on variance in performance. They evaluate their method on SOTOPIA and SOTOPIA-Hard benchmarks and shows a 7.32% improvement in goal completion success over the GPT-4o baseline.

**Strengths:**

- The design is explicitly grounded in the Theory of Planned Behavior (TPB)
- The reward provide adequate signals to help improve performance on the SOTOPIA benchmark
- Fine-tuned Qwen2.5-7B has better performance than GPT-4o baseline

**Weaknesses:**

- There are several related works that perform hierarchy with LLMs, including different versions of ReACT that are goal-conditioned. How does your method compare to them?
- The comparisons in Table 1 appear incomplete. It is not fair to directly compare the performance of GPT-4o with Qwen2.5-7B-Instruct (ReACT), as they differ significantly in scale and capability. How does GPT-4o perform when equipped with the ReACT framework? Does it outperform your proposed method? I recommend rerunning the second part of Table 1 using all algorithms on the same, strongest model backbone to ensure a fair and controlled comparison.
- From my understanding, your algorithm relies heavily on multiple LLM evaluators which could incur high costs. Would this be an issue in more long-horizon tasks?

**Questions:**

- How long are the dialogues generated? Is this truly testing long-horizon capabilities?
- I am not sure if the three stages of Thinking, Strategy, and response is necessarily hierarchical, and seems more so like planning which has been done. Additionally, the linearization of the Hierarchical Reinforcement Learning (HRL) structure is a bit confusing.
- Did you consider other social interaction benchmarks?

---

> ### Author Response · Authors · 2025-11-21
> **Response to rewiewer hKzL [1/2]**
>
> Thanks for your constructive feedback. We respond to your questions one by one:
> >W1 – Fair comparison against related hierarchical methods (e.g., ReAct) and model scales.
>
> We additionally include PoAct [1] in the comparison, and the updated results are:
>
> |Method|SOTOPIA GOAL|OVERALL|Hard GOAL|Hard OVERALL|AVG|
> |-|-|-|-|-|-|
> |vanilla|6.71|3.13|5.90|2.90|4.66|
> |ReAct|6.57|3.07|5.54|2.87|4.51|
> |PoAct|6.65|3.13|5.87|2.93|4.64|
> |TSR|**6.74**|**3.17**|**5.91**|**2.97**|**4.68**|
>
>
> TSR outperforms all test-time–scaling baselines. Prior studies have shown that such test-time reasoning often fails to help in role-playing or social-interaction tasks due to inconsistency and framework interference [2,3]. In our work, TSR is designed to model human mental processes: the Thinking–Strategy–Response decomposition exposes intermediate reasoning steps so that RL can align them or use them as additional reward signals. **Its goal is not only to improve performance, but also to produce interpretable reasoning traces that support stable hierarchical training.**
>
>
>
>
>
>
>
> >w2 - The comparisons in Table 1 appear incomplete.
>
> We additionally applied both ReAct and TSR to GPT-4o on SOTOPIA / SOTOPIA-Hard:
>
> |Method|SOTOPIA GOAL|SOTOPIA OVERALL|SOTOPIA-Hard GOAL|SOTOPIA-Hard OVERALL|AVG|
> |-|-|-|-|-|-|
> |gpt-4o|8.19|3.76|6.97|3.46|5.60|
> |gpt-4o w/ ReAct|8.22|3.70|6.88|3.44|5.56|
> |gpt-4o w/ TSR|8.20|3.81|6.92|3.52|5.61|
> |Qwen2.5-7B|6.71|3.13|5.90|2.90|4.66|
> |Qwen2.5-7B w/ ReAct|6.57|3.07|5.54|2.87|4.51|
> |Qwen2.5-7B w/ TSR|6.74|3.17|5.91|2.97|4.68|
>
> For GPT-4o, both ReAct and TSR bring no clear gains and sometimes slightly hurt performance. We hypothesize this is because GPT-4o already has **strong built-in chain-of-thought behavior**, and adding external test-time scaffolding can interfere with its internal reasoning rather than help.
>
> >I recommend rerunning the second part of Table 1 using all algorithms on the same, strongest model backbone to ensure a fair and controlled comparison.
>
> Our main goal is to train **Qwen2.5-7B**, which represents the realistic upper bound that *most researchers can actually train*. Training closed-source commercial models like GPT-4o is not feasible. We report GPT-4o’s behavior only to show method validity, while all training-based comparisons are conducted fairly and consistently on Qwen2.5-7B.
>
>
>
> >W3 – Cost of multiple LLM evaluators.
>
>
> We address the cost concern as follows.
> (1) In long-horizon settings, not every step is used for training. We apply a data-filtering procedure (Appendix A.2) so only high-quality turns are kept, which substantially reduces the number of evaluator calls.
> (2) For this work, we use a commercial LLM evaluator mainly to keep the pipeline simple and reproducible. In real large-scale deployments, it is common to replace the evaluator with a lightweight open-source reward model (need extra sft procedure), in which case the additional cost becomes negligible.
> (3) We also adopt list-wise scoring, allowing the evaluator to produce multiple scores in a single call. This reduces token usage and improves efficiency. We plan to further explore more efficient evaluation strategies in future work.
>
> >Q1 – Do experiments really stress long-horizon dialogues?
>
> Thanks for raising the question about “long-horizon” capability. Here we clarify that the core of social-interaction evaluation is not long-horizon planning in the classical sense. As stated in the SOTOPIA paper, dialogues are capped at 20 turns and are designed to assess goal achievement and relationship management “within a relatively short timespan in single episodes” [4]. Thus, if long-horizon refers to extended, multi-stage, long-dependency task chains, SOTOPIA is not intended to test that ability.
>
> Our work also does not target long-horizon reasoning per se. The focus is on modeling strategic social behaviors—thinking, planning, and responding—within bounded multi-turn interactions. Over 60% of generated dialogues exceed 15 turns (examples provided in Appendix L), which sufficiently matches the setting of SOTOPIA, but the goal of our method is **social-behavior alignment rather than solving long-horizon planning problems**.
>
> ----
> [1] *PoAct: Policy and Action Dual-Control Agent for Generalized Applications.* https://arxiv.org/abs/2501.07054
>
> [2] *Reasoning Does Not Necessarily Improve Role-Playing Ability.* https://arxiv.org/pdf/2502.16940
>
> [3] *EPO: Explicit Policy Optimization for Strategic Reasoning in LLMs via Reinforcement Learning.* https://arxiv.org/pdf/2502.12486
>
> [4] *SOTOPIA: Interactive Evaluation for Social Intelligence in Language Agents.* https://arxiv.org/abs/2310.11667

---

> ### Author Response · Authors · 2025-11-21
> **Response to rewiewer hKzL [2/2]**
>
> >Q2 – Is the Think–Strategy–Response pipeline genuinely hierarchical and how is “linearized HRL” different from prior planning?
>
> We thank the reviewer for pointing out the potential confusion around whether TSR is “just planning” or resembles existing plan-and-act pipelines. We have updated the paper to clarify the following.
>
> TSR is not a conventional planning module. Planning frameworks typically merge reasoning and action selection into a single chain-of-thought, where speculation, heuristics, meta-reasoning, and action candidates are all intertwined. In contrast, TSR explicitly separates *cognitive reasoning* from *behavioral intention formation*.
>
> As detailed in Section 3 and illustrated in Figure 1, TSR is a hierarchy of abstraction:
>
> - **Thinking** captures latent cognitive factors—speculation, inference, social norms, control beliefs—that shape how an agent interprets the interaction.
> - **Strategy** turns this cognition into an explicit, human-interpretable behavioral intention.
> - **Response** is the surface-level utterance conditioned on that intention.
>
> This separation is not only for interpretability. It is essential for our training algorithm: LHRL-VGR relies on stage-specific reward routing and variance-gated assignment, which require stable intermediate reasoning traces (especially strategies). Without exposing these intermediate signals, hierarchical credit assignment becomes unreliable.
>
> (2) Why we “linearize” HRL in this dialogue setting.
> Classical HRL assumes that sub-policies operate over extended temporal horizons. In contrast, each dialogue turn must immediately produce one utterance (one action). Therefore, our HRL structure is linearized into a per-turn pipeline:
>
> * the high-level stage (strategy) produces an intention for this turn only,
>
> * the low-level stage executes it to produce the utterance.
>
> This linearization is not an approximation of HRL but a necessary adaptation to the turn-based nature of dialogues. Figure 2 and Section 4 further show that the two levels still receive separated and level-appropriate reward signals (strategy content vs. variance-gated action reward), preserving the spirit of hierarchical credit assignment even within a linear generation pipeline.
>
> Our ablations (Table 2) and training-order experiments (Figure 3) show these intermediate signals are necessary for stable optimization.
>
> We hope this clarifies that TSR is not equivalent to existing planning methods, but a structured hierarchy that supports interpretable reasoning and reliable hierarchical reinforcement learning.
>
>
>
> >Q3 – Other benchmarks.
>
> Without any additional training, we directly evaluated our social agent (LHRL-VGR) on the DCN benchmark [5], which focuses on debt-collection negotiations grounded in real-world scenarios. The results are:
>
> |Model|CRI|DHI|CCI|
> |-|-|-|-|
> |Qwen-2.5-7B|0.732|**0.793**|0.743|
> |GPT-4o|**0.844**|0.580|0.774|
> |**LHRL-VGR**|0.834|0.665|**0.795**|
>
> Our method again outperforms GPT-4o on the overall CCI metric, suggesting that LHRL-VGR generalizes well to goal-oriented social-interaction tasks beyond the original benchmark.
>
> ----
>
> [5] *Debt Collection Negotiations with Large Language Models: An Evaluation System and Optimizing Decision Making with Multi-Agent.* https://arxiv.org/pdf/2502.18228

---

### Official Review · Reviewer_Tbqv · 2025-11-04

**Soundness:** 3
**Presentation:** 3
**Contribution:** 3
**Rating:** 6
**Confidence:** 4

**Summary:**

The paper proposes the TSR framework, a hierarchical algorithm that decomposes actions in social dialogue into high-level thinking and planning, and low-level responses. The key contribution is in jointly training high-level strategy and low-level response using LLM-evaluated rewards, using novel variance-gating to dynamically assign the reward of responses. The method achieves strong performance in the SOTOPIA benchmark.

**Strengths:**

1. The authors propose a principled framework for training hierarchical dialogue agents using custom rewards to jointly train both high-level planning and low-level execution. This is a clear improvement beyond traditional approaches that train using a single reward signal.

2. The variance-gated reward mechanism is novel, and addresses the challenge of noisy LLM evaluators when using LLM-as-a-judge for training.

3. The method achieves strong empirical performance on challenging benchmarks.

**Weaknesses:**

1. A major downside is the complete  reliance on LLM evaluation for reward signals. This makes it unclear whether potential biases in the evaluator will limit the generalizability of the approach to other social dialogue benchmarks.

2. It is unclear how important the non-repetition reward is, as it seems like mostly a "hack" to avoid repetitive behavior. It would be interesting to see an ablation with this reward removed from the objective, to understand if it is truly necessary for performance.

3. The method itself should be noticeably more complex than evaluated baselines, as to my knowledge, it is the only one that performs 2-stage training. It would be helpful if the authors made clear what the additional overhead is during training and inference.

**Questions:**

1. Have the authors considered using different policy models for high-level strategy and low-level response generation? It would be interesting to see if that has any impact on final performance.

2. The authors use a scaling function on the goal-completion reward. How important is this scaling, i.e. would the method still work well with standard normalization or just using the raw difference?

---

> ### Author Response · Authors · 2025-11-21
> **Response to rewiewer Tbqv [1/2]**
>
> Thanks for your constructive feedback. We respond to your questions one by one:
> > W1 – Reliance on LLM rewards and potential evaluator bias.
>
> 1. **Validation of our reward design:** To further clarify the rigor of our reward design, we added Appendix L (marked in red in the revision), where we spell out the detailed evaluation criteria for each reward dimension.
>
> To check whether GPT-4o can reliably serve as our reward evaluator, we randomly sampled 100 episodes and asked three human experts (all graduate students in computational linguistics) to act as “human judges.” Each episode was annotated by two humans independently, and disagreements were resolved by a third annotator. We then compared the human scores with GPT-4o’s scores. The Pearson correlation results are:
>
> |Reward dim.|Predicted Outcome|Content Completeness|Goal Completion|Strategy-followed|
> |-|-|-|-|-|
> |**Correlation (r)**|0.59|0.87|0.74|0.67|
>
> All correlations are above 0.5, and all p-values satisfy **p ≤ 0.01**, showing that GPT-4o aligns well with human judgment across all reward dimensions.
>
>
> 2. **Cross-benchmark generalization:** Without any additional training, we directly evaluated our social agent (LHRL-VGR) on the DCN benchmark [1], which focuses on debt-collection negotiations grounded in real-world scenarios. The results are:
>
> |Model|CRI|DHI|CCI|
> |-|-|-|-|
> |Qwen-2.5-7B|0.732|**0.793**|0.743|
> |GPT-4o|**0.844**|0.580|0.774|
> |**LHRL-VGR**|0.834|0.665|**0.795**|
>
>
> Our method again outperforms GPT-4o on the overall CCI metric, suggesting that LHRL-VGR generalizes well to goal-oriented social-interaction tasks beyond the original benchmark.
>
> > W2 – Importance of the non-repetition reward.
>
> Table 2 reports the ablation results for the non-repetition reward:
>
> |Method|GOAL|REL|OverView|Rep. rate|
> |-|-|-|-|-|
> |LHRL-VGR|7.63|3.40|3.69|0.11|
> |w/o r_rep|7.59|3.37|3.53|0.69|
>
> Removing r_rep hurts performance across all metrics, with the largest drop on the OverView score (3.69 → 3.53). We also measured how often the model repeated previous turns by sampling 200 dialogues. A dialogue was counted as “repetitive’’ if any utterance repeated an earlier one. The repetition rate jumps from **0.11 to 0.69** without r_rep, showing that this reward is doing more than a simple “hack’’—it substantially prevents degenerate looping behavior.
>
> For clarity, Appendix L (Tables 13–14) includes an example dialogue comparing outputs with and without the non-repetition reward, along with a brief analysis.
>
>
> > W3 – Additional complexity of the two-stage training.
>
> The main difference in complexity comes from inference-time reasoning rather than from the training loop itself.
> During RL training, each sample still goes through one forward–backward pass, so the per-sample compute is unchanged.
> The extra cost mainly comes from (1) longer reasoning traces during inference and (2) the GPT-4o calls we use for reward annotation.
>
> Below is the average token usage per episode:
>
> |Method|Tokens used in inference|Tokens used for RL reward annotation (GPT-4o)|
> |-|-|-|
> |direct|87|–|
> |LHRL-VGR|685|462|
>
>
> Most of the total training time comes from calling the GPT-4o API for reward annotation. This is a design choice made to keep the research pipeline clean and consistent. In future work, we plan to train a much smaller reward model to replace GPT-4o for annotation, which will significantly reduce the overhead.
>
> ----
> [1] *Debt Collection Negotiations with Large Language Models: An Evaluation System and Optimizing Decision Making with Multi-Agent.* https://arxiv.org/pdf/2502.18228

---

> ### Author Response · Authors · 2025-11-21
> **Response to rewiewer Tbqv [2/2]**
>
> > Q1 – Using different policy models for high-/low-level behaviors.
>
> The results below show that using different policy models for high-level and low-level reasoning does affect performance. Among all combinations, HRLDR achieves the best scores:
>
> |high-\low-|Qwen2.5|gpt-4o|Goal-RL|HRLDR|
> |-|-|-|-|-|
> |**Qwen2.5**|5.91|6.30|6.27|6.84|
> |**gpt-4o**|6.08|6.90|7.04|7.13|
> |**Goal-RL**|6.57|7.10|7.23|7.20|
> |**HRLDR**|7.03|7.15|7.20|**7.65**|
>
> We can see that stronger high-level planners generally lead to better low-level responses, and using HRLDR for the high-level stage yields the highest overall score. The high-level planner determines strategy quality and global coherence, while the low-level layer mainly controls execution quality.
>
>
>
>
> > Q2 – Necessity of the goal-reward scaling.
>
> The goal-completion reward uses a boundary-aware scaling function to stabilize training.
> Raw score differences from the LLM evaluator can vary depending on how close the dialogue is to the goal boundaries.
> The scaling adjusts the magnitude of positive/negative progress so that rewards remain well-behaved and mapped into a consistent range, which improves learning stability.
> When we remove this scaling and use raw differences, performance drops:
>
> |Method|GOAL|REL|OverView|
> |-|-|-|-|
> |LHRL-VGR|7.63|3.40|3.69|
> |w/o reward scaling|7.45|3.32|3.55|
>
> This scaling design follows the goal reward formulation used in AMPO [2], and we have added a reference note in Section 4 (marked in red) to clarify this connection.
>
> ----
> [2] *Adaptive Thinking via Mode Policy Optimization for Social Language Agents*. https://arxiv.org/abs/2505.02156

---

### Author Response · Authors · 2025-12-03
**Final Summary**

Dear ACs/SACs/PCs

**We sincerely thank you for your time and effort in managing the review process for our paper.** We provide a summary of the current status of our paper and the progress of our rebuttal.

---

## Strengths Recognized by Reviewers

Reviewers highlighted several strengths:

- **A principled hierarchical design (TSR) and a novel RL algorithm (LHRL-VGR).**
  Reviewers acknowledged the clear motivation, structured decomposition, and the variance-gated reward routing mechanism. (Tbqv, hKzL, xWGw)

- **Comprehensive experiments and strong empirical gains.**
  Multiple reviewers noted the performance improvements across SOTOPIA, SOTOPIA-Hard, and DCN. (Tbqv, hKzL)

- **Clear writing and presentation.**
  Reviewers found the method explanation and reward motivation easy to follow. (Tbqv, hKzL)

---

## Addressed Concerns

All concerns raised during review were fully addressed during the rebuttal:

- **Reward interpretability (V13c).**
  We added matched-turn case studies showing how each reward component changes behavior, and clarified their roles in preventing goal drift, rigidity, and repetition.

- **Reliability of LLM-as-judge rewards (Tbqv, xWGw).**
  We added a human–model agreement study (100 samples, triple annotation) and cited SOTOPIA’s human-LLM correlations.

- **Comparison fairness with ReAct/PoAct and model scales (hKzL).**
  We added ReAct/PoAct baselines and tested both methods on GPT-4o, showing that test-time scaffolding does not benefit strong commercial models. All trainable comparisons remain on Qwen2.5-7B for fairness.

- **Cost and efficiency concerns (hKzL, xWGw).**
  We documented token costs and clarified that commercial evaluators can be replaced by compact reward models in deployment.

- **Ethics considerations (xWGw).**
  We expanded the ethics statement with concrete discussions on misuse risks and mitigation.

- **Incremental contribution concern (V13c).**
  We added additional experiments showing:
  (1) Stage-1 strategies alone outperform SOTOPIA-RL when used as rewards;
  (2) Our co-trained reasoning stage improves baselines such as AMPO;
  (3) Our learned strategies generalize to other models and tasks.

---
**Before the OpenReview information leak incident**, we had already received responses from two reviewers:

- Reviewer xWGw acknowledged that we addressed most of his concerns and **raised his score from 6 to 8**.
- Reviewer V13c engaged with us in two further rounds of discussion. Although we respectfully disagreed with some of his characterizations, we responded point-by-point by grounding our explanations directly in the case-study examples he highlighted, clarifying the qualitative interpretations he questioned, and providing additional ablation results and cross-model experiments to substantiate the effects of each reward component.

## Final Remarks

All reviewer concerns were resolved during the rebuttal, and no new issues were raised afterward. We deeply appreciate the reviewers' and AC’s efforts and the opportunity to improve our work.

**Best regards,**
*Submission7220 Authors*

---

### Meta-Review · Area_Chair_PEsG · 2026-01-11

**Summary:**

This paper introduces Think-Strategy-Response (TSR) framework and a corresponding RL optimization method (LHRL-VGR) for enhancing the social intelligence of LLMs. While the reviewers acknowledged the strong empirical performance and the motivation for hierarchical decomposition, the decision for rejection is driven by persistent concerns regarding the theoretical justification and design choices of the core contribution. For example, the Variance-Gated Reward mechanism was criticized for being an arbitrary heuristic without sufficient grounding, and the interpretability of the reward components remains a point of contention. Despite a thorough rebuttal that addressed many empirical concerns, the fundamental skepticism about the principled nature of the method prevents acceptance.

**Reviewer Concerns:**

**Concerns addressed by rebuttal**
- Reviewers xWGw, Tbqv questioned the reliability of using LLMs for evaluation and reward signals. The authors addressed this by adding a human-model agreement study (100 episodes) reporting reasonable correlations.
- The concern about the necessity of non-repetition reward (Reviewer Tbqv) was addressed with an ablation study showing a performance drop without it, justifying its inclusion as a necessary component for multi-turn dialogue stability.


**Outstanding concerns**
- In response to fairness concerns (Reviewer hKzL), the authors applied TSR to GPT-4o and reported no gains or slight performance drops, attributing this to interference with GPT-4o's internal reasoning. This defense is unconvincing. Unlike specialized reasoning models (e.g., OpenAI o1), GPT-4o is a generalist model. The fact that TSR hinders a capable model suggests the framework imposes rigid, suboptimal constraints rather than providing a general intelligence gain. It implies the method can be effective only for weaker models (e.g., Qwen-7B) to cover their deficiencies, lacking utility for state-of-the-art systems.
- Reviewer V13c consistently questioned the design of the specific form of rewards (e.g., non-repetition, strategy adherence, variance-gated reward), arguing that these choices lack a clear derivation from established RL principles and appear ad-hoc. The rebuttal provided empirical ablations but failed to offer a theoretical basis for why variance-gating is the correct mechanism, leaving the reviewer unconvinced. The reviewer explicitly stated that the rebuttal failed to convince them of the non-arbitrary nature of the design.
- Reviewer V13c challenged whether the qualitative examples provide sufficient supporting evidence for the method's claims. While the authors corrected earlier issues (including the use of placeholder descriptions), the reviewer's core skepticism remains: it is unclear whether the observed behaviors reflect genuine strategic reasoning induced by the proposed reward structure, or merely overfitting to the LLM-based reward signals.
- The decomposition of Think-Strategy-Action was viewed as a standard Chain-of-Thought (CoT) pattern applied to HRL. The reviewer argued that the technical contribution is incremental, with the social intelligence aspect being largely handled by the pre-trained LLM and prompt engineering rather than the RL algorithm itself.

**Reviewer Scores:**

- Reviewer xWGw (8): Score raised from 6 to 8. This reviewer valued the empirical performance and being satisfied with the human agreement study. They explicitly stated they would update their score upward.
- Reviewer Tbqv (6): Likely to remain 6. The rebuttal clarified how much it costs, but not necessarily why it is worth it given the limitations.
- Reviewer hKzL (4): Likely to remain 4. Although the authors performed the requested experiments (applying TSR to GPT-4o), the results showed no performance gain, effectively validating the reviewer's initial skepticism about the method's utility on stronger models.
- Reviewer V13c (2): Likely to remain 2. This reviewer provided the most critical assessment, refusing to accept the arbitrary reward design. Their skepticism regarding the principled nature of the contribution remains the primary hurdle for this paper.

---

### Decision · Program_Chairs · 2026-01-26

Reject